# Low-Rank Orthogonalization for Large-Scale Matrix Optimization with Applications to Foundation Model Training

## Abstract

Neural network (NN) training is inherently a large-scale matrix optimization problem, yet the matrix structure of NN parameters has long been overlooked. Recently, the optimizer Muon (Jordan et al., 2024b), which explicitly exploits this structure, has gained significant attention for its strong performance in foundation model training. A key component contributing to Muon's success is matrix orthogonalization. In this paper, we propose *low-rank orthogonalization*, which performs orthogonalization by leveraging the low-rank nature of gradients during NN training. Building on this, we introduce low-rank matrix-signed gradient descent (MSGD) and a low-rank variant of Muon. Numerical experiments demonstrate the advantages of low-rank orthogonalization: low-rank Muon generally matches or improves upon vanilla Muon on the GPT-2 and LLaMA pretraining tasks, with clearer improvements observed for relatively larger models. Theoretically, we establish the iteration complexity of low-rank MSGD for finding an approximate stationary solution, and the iteration complexity of low-rank Muon for finding an approximate stochastic stationary solution under heavy-tailed noise. The code to reproduce our numerical experiments is available at https://anonymous.4open.science/r/Low-rank-Muon-75C6/.

**Keywords:** Orthogonalization, Muon, foundation model training, iteration complexity, heavy-tailed noise

## 1 Introduction

Training neural networks (NNs) (LeCun et al., 2015), particularly recent foundation models, has consistently posed challenging large-scale optimization problems. Over the past decade, NN training has been dominated by vector-variate optimization methods—including SGD (Bottou, 2010), AdaGrad (Duchi et al., 2011), RMSprop (Hinton et al., 2012), Adadelta (Zeiler, 2012), Adam (Kingma & Ba, 2015), and AdamW (Loshchilov & Hutter, 2019). Nonetheless, these methods disregard the inherent matrix structure of NN parameters—such as those in multi-layer perceptrons (Rosenblatt, 1958), convolutional layers (LeCun et al., 1989), and the query, key, and value projections in attention mechanisms (Vaswani et al., 2017).

Recently, a shift has occurred: optimization methods that exploit matrix structure are receiving increasing attention and have begun to demonstrate strong performance in foundation model training (Jordan et al., 2024b; Pethick et al., 2025). These methods focus on solving the matrix optimization problem:

$$\min_{X \in \mathbb{R}^{m \times n}} f(X). \tag{1}$$

In particular, Shampoo, as developed in (Anil et al., 2020; Gupta et al., 2018), applies left and right preconditioning matrices and updates them as follows:

$$X^{k+1} = X^k - \eta_k (L^k)^{-1/4} G^k (R^k)^{-1/4}, \tag{2}$$

where $\eta_k > 0$ is the step size, $G^k \in \mathbb{R}^{m \times n}$ denotes the (stochastic) gradient of $f$ at $X^k$, and $L^k \in \mathbb{R}^{m \times m}$ and $R^k \in \mathbb{R}^{n \times n}$ are the left and right preconditioning matrices, respectively. Shampoo updates the left and right preconditioners $\{L^k\}$ and $\{R^k\}$ using the second-order statistics of the accumulated gradients, similarly

to AdaGrad, and has shown comparable performance to popular vector-variate optimizers such as Adam and AdamW on foundation model training. Following Shampoo, other matrix-variate optimizers that use two-sided preconditioners, such as CASPR (Duvvuri et al., 2024) and SOAP (Vyas et al., 2025), were also developed. A preconditioned Riemannian gradient descent method was developed in (Bian et al., 2024) for low-rank matrix recovery, which adopts only the diagonal part of the Shampoo preconditioners. Moreover, one-sided preconditioned variants of Shampoo were developed in (An et al., 2025; Xie et al., 2025).

In addition to Shampoo and its variants, another matrix-variate optimizer, Muon (Jordan et al., 2024b), has attracted significant attention for outperforming standard optimizers such as Adam and AdamW in foundation model training (AI et al., 2025). At each iteration, Muon performs the update:

$$M^k = (1 - \theta_{k-1})M^{k-1} + \theta_{k-1}G(X^k; \xi^k), \quad X^{k+1} = X^k - \eta_k \mathrm{msgn}(M^k), \tag{3}$$

where $G(\cdot; \xi)$ denotes the stochastic gradient of $f(\cdot)$, and $\mathrm{msgn}(M^k) = U^k(V^k)^T$ denotes the matrix sign of $M^k$, with $U^k$ and $V^k$ containing the left and right singular vectors of $M^k$, respectively. The matrix sign computation is often referred to as matrix orthogonalization, because calculating $\mathrm{msgn}(M)$ is equivalent to finding the (semi-)orthogonal matrix closest to $M$ with respect to the Frobenius norm (see, e.g., (Bernstein & Newhouse, 2024, Proposition 4)). Muon's empirical success has sparked significant research interest, including efforts to understand its relationship with other algorithms, establish its convergence guarantees, and propose new variants (see, e.g., (An et al., 2025; Cesista, 2025; Chen et al., 2025; Glentis et al., 2025; Kovalev, 2025; Lau et al., 2025; Li & Hong, 2025; Liu et al., 2025; Ma et al., 2024; Pethick et al., 2025; Riabinin et al., 2025; Sato et al., 2025; Sfyraki & Wang, 2025; Shen et al., 2025; He et al., 2025b; Bogachev et al., 2026; Ma et al., 2026)). A popular interpretation of Muon is from the perspective of a linear minimization oracle with respect to the spectral norm (e.g., see (Bernstein & Newhouse, 2024; Cesista, 2025; Glentis et al., 2025; Kovalev, 2025; Lau et al., 2025; Riabinin et al., 2025)). That is, the matrix sign computation in (3) can be recast as $-\mathrm{msgn}(M^k) = \arg\min_{\|\Delta\| \le 1}\{\langle M^k, \Delta \rangle\}$, where $\|\cdot\|$ denotes spectral norm. Based on this interpretation, algorithmic designs leveraging general matrix-induced norms $\|\cdot\|_{p \to q}$ have been discussed in (Bernstein & Newhouse, 2024; Cesista, 2025; Glentis et al., 2025; Riabinin et al., 2025). In addition, Muon is also connected to earlier algorithms and can be viewed as a special case of Shampoo, despite not explicitly using preconditioners. As discussed in (Jordan et al., 2024b), by taking $L^k = G^k(G^k)^T$ and $R^k = (G^k)^T G^k$ in (2), the Shampoo updates reduce to the matrix-signed update: $X^{k+1} = X^k - \eta_k \mathrm{msgn}(G^k)$. Furthermore, convergence guarantees for Muon have been extensively studied (e.g., see (An et al., 2025; Chen et al., 2025; Kovalev, 2025; Li & Hong, 2025; Riabinin et al., 2025; Sato et al., 2025; Sfyraki & Wang, 2025; Shen et al., 2025)), and numerous new variants—such as SWAN (Ma et al., 2024), Scion (Pethick et al., 2025), Gluon (Riabinin et al., 2025), PolarGrad (Lau et al., 2025), Dion (Ahn & Xu, 2025; Ahn et al., 2025), and AdaMuon (Si et al., 2025)—have been proposed. Notably, unlike existing analyses of Muon-type algorithms, our analysis considers heavy-tailed noise (see Assumption 2) while allowing inexact computation of the matrix sign operation (see Algorithm 4).

Beyond applying orthogonalization, a key innovation of Muon is its use of a GPU-friendly method—Newton-Schulz iterations (typically with five steps)—to perform inexact matrix orthogonalization, making it well-suited for modern foundation model training. In fact, several earlier methods, including spectral gradient descent (Carlson et al., 2015a;b;c) and orthogonalized gradient descent (Tuddenham et al., 2022), have already adopted SVD-based orthogonalization for matrix optimization problems. However, since SVD is not computationally efficient on GPUs when training large-scale neural networks, these methods fail to scale to foundation model training.

Inspired by Muon and its orthogonalization subroutine, we aim to develop a faster and more lightweight orthogonalization method to further enhance Muon and its variants. Specifically, our design leverages the widely observed phenomenon that the gradient matrices of NN parameters are often low-rank (see, e.g., (Hao et al., 2024; Hu et al., 2022; Malladi et al., 2023; Zhao et al., 2024)). To exploit this low-rank property, we propose performing low-rank orthogonalization by incorporating well-known low-rank matrix approximation techniques (Drineas et al., 2006a; Halko et al., 2011). Our approach first constructs a low-rank projection of the gradient matrix using QR decomposition on a sketched matrix, and then performs orthogonalization on the projected matrix by leveraging its structure. Our proposed low-rank orthogonalization offers two main advantages over existing orthogonalization methods:

1. **Computational Efficiency:** Traditional orthogonalization can be seen as computing the polar factor of a given full matrix. To exploit the low-rank property, our low-rank orthogonalization first computes the $Q$ factor of a smaller sketched matrix, and then computes the polar factor of a projected matrix constructed using the $Q$ factor. Since both the QR decomposition and polar decomposition are performed on much smaller matrices, our low-rank approach enjoys substantial computational savings for large-scale problems.

2. **Noise Robustness:** In the presence of noise, singular vectors associated with small singular values often vary significantly, leading to instability when directly applying orthogonalization to the full matrix. To circumvent this instability, our low-rank orthogonalization method clips these vectors to eliminate unreliable estimates, thereby stabilizing the orthogonalization process and yielding a robust estimate of the matrix sign.

These advantages will be illustrated in detail in Sections 3.1 and 4. Based on low-rank orthogonalization, we develop low-rank matrix-signed gradient descent (MSGD) and a low-rank variant of Muon. We also establish their complexity guarantees under mild assumptions.

Our main contributions are highlighted below.

- We propose *low-rank orthogonalization* that can be readily incorporated into matrix-variate optimization algorithms such as Muon. It can be efficiently executed on GPUs and serves as a lightweight substitute for traditional orthogonalization methods that are directly applied to the full matrix, such as Newton-Schulz iterations.

- Under mild assumptions, we establish the iteration complexity of low-rank MSGD and a low-rank variant of Muon. To the best of our knowledge, our results provide the first complexity guarantees for a broad class of inexact Muon-type algorithms, including vanilla Muon, under heavy-tailed noise.

The remainder of this paper is organized as follows. In Section 2, we introduce the notation and assumptions used throughout the paper. In Section 3, we propose low-rank orthogonalization and, based on it, develop low-rank MSGD and low-rank Muon. In Section 4, we present numerical results. Finally, we provide the proofs of the main results in Section B.

## 2   Notation and Assumptions

Throughout this paper, we use $\mathbb{R}^{m \times n}$ to denote the Euclidean space of $m \times n$ real matrices, and $\mathbb{Z}_+$ to denote the set of all nonnegative integers. We use $\|\cdot\|$ to denote the Euclidean norm of a vector or the spectral norm of a matrix; $\|\cdot\|_*$ and $\|\cdot\|_F$ to denote the nuclear norm and the Frobenius norm of a matrix, respectively; and $\langle \cdot, \cdot \rangle$ to denote the trace inner product for matrices. For any $M \in \mathbb{R}^{m \times n}$, we use $\mathrm{rank}(M)$ to denote its rank, and $[M]_k$ to denote its best rank-$k$ approximation with respect to $\|\cdot\|_F$. We define the matrix sign of any nonzero matrix $M \in \mathbb{R}^{m \times n}$ as $\mathrm{msgn}(M) = UV^T$, where $U \in \mathbb{R}^{m \times r}$ and $V \in \mathbb{R}^{n \times r}$ are column-orthogonal matrices obtained from the reduced SVD of $M$. We let $\varrho := \min\{m, n\}$. In addition, we use $\widetilde{\mathcal{O}}(\cdot)$ to denote $\mathcal{O}(\cdot)$ with logarithmic factors omitted.

We now make the following assumption throughout this paper.

**Assumption 1.** (a) *There exists a finite $f_{\mathrm{low}}$ such that $f(X) \geq f_{\mathrm{low}}$ for all $X \in \mathbb{R}^{m \times n}$.*

(b) *There exists an $L_* > 0$ such that $\|\nabla f(X) - \nabla f(Y)\|_* \leq L_* \|X - Y\|$ for all $X, Y \in \mathbb{R}^{m \times n}$.*

Assumption 1(a) is standard. Assumption 1(b) is natural in the analysis of Muon-type algorithms (e.g., see (Chen et al., 2025; Riabinin et al., 2025; Shen et al., 2025)). It follows from Assumption 1(b) that

$$f(Y) \leq f(X) + \langle \nabla f(X), Y - X \rangle + \frac{L_*}{2} \|Y - X\|^2 \qquad \forall X, Y \in \mathbb{R}^{m \times n}. \tag{4}$$

We next provide a definition for approximate stationary points of problem (1).

**Definition 1.** *For any $\epsilon \in (0,1)$, we say that $X \in \mathbb{R}^{m \times n}$ is an $\epsilon$-nuclear norm stationary point (NSP) of problem (1) if it satisfies $\|\nabla f(X)\|_* \leq \epsilon$, and that it is an $\epsilon$-stochastic nuclear norm stationary point (SNSP) of problem (1) if it satisfies $\mathbb{E}[\|\nabla f(X)\|_*] \leq \epsilon$. Here, the expectation is taken with respect to all randomness in the algorithm and in the generation of the solution.*

## 3 Matrix Optimization with Low-Rank Orthogonalization

In this section, we propose matrix optimization algorithms with low-rank orthogonalization for solving (1). In particular, we first propose low-rank orthogonalization in Section 3.1. Then, we propose low-rank MSGD in Section 3.2, and a low-rank variant of Muon in Section 3.3.

### 3.1 Low-Rank Orthogonalization

Orthogonalization techniques have attracted increasing attention in recent optimizer designs, as it has shown strong empirical performance in foundation model training (e.g., see (Bernstein & Newhouse, 2024; Jordan et al., 2024b; Lau et al., 2025; Tuddenham et al., 2022)). In this subsection, we develop a low-rank orthogonalization method, leveraging low-rank matrix approximation techniques, that serves as a lightweight substitute for the existing orthogonalization subroutines used in matrix-variate optimizers.

---
**Algorithm 1** A Low-Rank Orthogonalization Method

---
**Input:** matrix $M \in \mathbb{R}^{m \times n}$, rank trial $r \in \mathbb{Z}_+ \cap [1, \varrho]$.
**Output:** approximate matrix sign $M_O \in \mathbb{R}^{m \times n}$.
Draw a Gaussian random matrix $G \in \mathbb{R}^{n \times r}$.
Perform a QR decomposition on $MG$ to obtain a column-orthogonal Q factor $Q \in \mathbb{R}^{m \times r}$.
Return $M_O = Q\mathrm{msgn}(Q^T M)$. (On GPUs, $\mathrm{msgn}(Q^T M)$ is recommended to be estimated via Newton-Schulz iterations.)

---

Specifically, our low-rank orthogonalization method, presented in Algorithm 1, is based on Gaussian sketching (Halko et al., 2011). This method first draws a Gaussian random matrix $G \in \mathbb{R}^{n \times r}$ with $r \ll \varrho$, and performs a QR decomposition on $MG$ to obtain a column-orthogonal Q factor $Q \in \mathbb{R}^{m \times r}$. Then, it computes $\mathrm{msgn}(Q^T M) \in \mathbb{R}^{r \times n}$ and returns $M_O = Q\mathrm{msgn}(Q^T M)$ as a low-rank approximation for $\mathrm{msgn}(M)$. As will be shown in Theorem 1, $M_O$ represents the matrix sign of $QQ^T M$, which is a low-rank approximation of $M$. Its proof is deferred to Section B.1.

**Theorem 1.** *Consider Algorithm 1 with inputs $M \in \mathbb{R}^{m \times n}$ and $r \in \mathbb{Z}_+ \cap [1, \varrho]$, where $\varrho := \min\{m, n\}$. Let $Q \in \mathbb{R}^{m \times r}$ be generated by Algorithm 1. Then, for any $r_*$ satisfying $2 \leq r_* \leq r - 2$, it holds that*

$$\mathbb{E}[\|(I - QQ^T)M\|_F] \leq \left(1 + \frac{r_*}{r - r_* - 1}\right)^{1/2} \|M - [M]_{r_*}\|_F. \tag{5}$$

*Moreover, we have $\mathrm{msgn}(QQ^T M) = Q\mathrm{msgn}(Q^T M)$.*

**Remark 1.** *The relation (5) is adapted from (Halko et al., 2011, Theorem 10.5), where additional guarantees— such as those involving different matrix norms and high-probability bounds—can also be found. In addition, low-rank matrix approximation based on column selection (e.g., see (Drineas et al., 2006a;b)) can be used to develop a low-rank orthogonalization method. However, since its approximation guarantee is more complicated than that of the Gaussian sketching-based approach, we defer the column-selection-based method to the supplementary materials.*

Next, we illustrate two major advantages of our low-rank orthogonalization method, namely, *computational efficiency* and *noise robustness*, through synthetic experiments on randomly generated matrices.

**Computational Efficiency** We compare the computation time on GPUs for calculating inexact matrix sign of high-dimensional matrices using Newton-Schulz iterations, low-rank orthogonalization based on Gaussian sketching (Algorithm 1) and column selection (see supplementary materials), and truncated SVD.

For each $n \in \{1000, 2000, 5000, 10000\}$, we generate 50 random matrices $M \in \mathbb{R}^{n \times n}$, with each entry drawn from the standard Gaussian distribution. We then apply all competing orthogonalization methods to estimate $\text{msgn}(M)$. We implement Newton-Schulz iterations as provided in Muon (Jordan et al., 2024b) with 5 iterations. For our low-rank orthogonalization methods, we set $r = 0.1n$ as the input rank parameter, use command `torch.linalg.qr` to compute the Q factor, and apply 5 iterations of Newton-Schulz scheme following Muon (Jordan et al., 2024b) to compute the matrix sign of $Q^T M$. In addition, we use command `torch.pca_lowrank` to efficient perform truncated SVD, setting the rank to $0.1n$ for truncation.

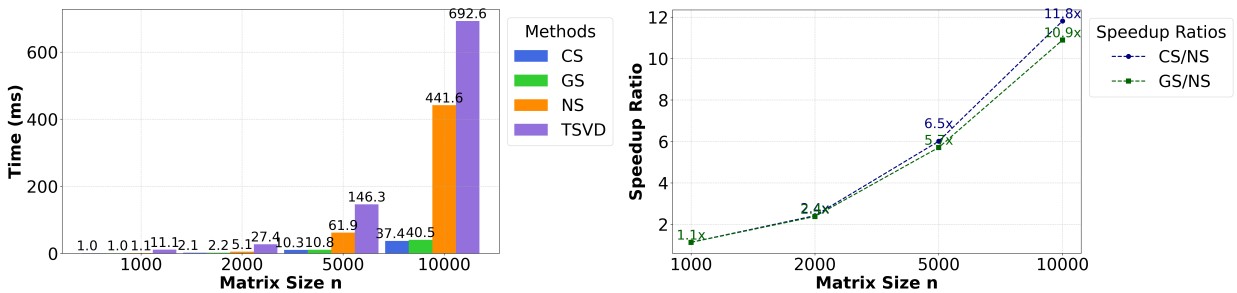

Figure 1: Left: Comparison of GPU computation time across Newton-Schulz iterations (NS), our low-rank orthogonalization with Gaussian sketching (GS) and column selection (CS), and truncated SVD (TSVD). Right: Speedup ratios of our low-rank orthogonalization methods compared to Newton-Schulz iterations.

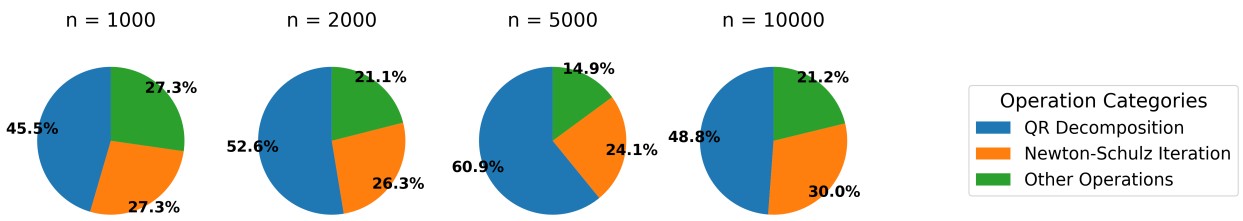

Figure 2: Time distribution for low-rank orthogonalization with Gaussian sketching, including the QR decomposition, the Newton-Schulz iterations for computing the matrix sign, and other computations.

We present a comparison of the computation time on GPUs for all competing methods in Figure 1, and the time distribution of our low-rank orthogonalization method with Gaussian sketching in Figure 2. From Figure 1, we observe that our low-rank orthogonalization method significantly reduces computation time compared to Newton-Schulz iterations and truncated SVD. Although both truncated SVD and our low-rank orthogonalization exploit low-rank structure for computation, truncated SVD is not well-suited for GPU environments and is therefore slower than the Newton-Schulz iterations. From Figure 2, we observe that in our low-rank orthogonalization method with Gaussian sketching, the QR decomposition accounts for approximately half of the total time, while the Newton-Schulz iterations and other operations (such as generating Gaussian random matrices and performing matrix multiplications) each make up roughly half of the remaining time.

**Noise Robustness**  In addition to reducing computation time, our low-rank orthogonalization methods also produce more robust estimates of the matrix sign for low-rank matrices in the presence of noise. This robustness is particularly important in foundation model training, where gradients often exhibit low-rank structure. We now compare the performance of Newton-Schulz iterations and our low-rank methods in estimating the matrix sign of noisy, low-rank matrices.

For each $n \in \{1000, 2000, 5000, 10000\}$, we first randomly generate 10 nearly low-rank matrices $M \in \mathbb{R}^{n \times n}$ following strategy: the top $0.1n$ singular values are set to 1, the remaining singular values are set to $10^{-4}$,

and the singular vectors are randomly generated orthogonal vectors. For each $M$, we generate 50 noise matrices $N \in \mathbb{R}^{n \times n}$, with each entry drawn from a Gaussian distribution with mean zero and standard deviation $\sigma \in \{0.001, 0.01, 0.1, 0.5, 1\}$, and construct noisy matrices $M_N = M + N$. We next apply Newton-Schulz iterations, and our low-rank orthogonalization method based on Gaussian sketching (Algorithm 1), using an input rank parameter $0.1n$, to estimate $\mathrm{msgn}(M_N)$. For each $M$, we compute the trace of the empirical covariance matrix of the estimates[1] of $\mathrm{msgn}(M_N)$ produced by both Newton-Schulz iterations and Algorithm 1.

To complement the variance-based robustness metric, we also include a comparison between the estimated and ground-truth matrix sign operator. For $n \in \{1000, 2000, 5000, 10000\}$, we generate a nearly low-rank matrix $M \in \mathbb{R}^{n \times n}$ in the same manner as above. We then estimate $\mathrm{msgn}(M)$ from the noisy observation $M_N = M + N$ using Gaussian sketching (Algorithm 1) and the full Newton-Schulz iterations, where $N \in \mathbb{R}^{n \times n}$ has independent entries drawn from a Gaussian distribution with mean zero and standard deviation $\sigma = 0.0005$. Let $\widehat{S}(M_N)$ denote the estimate obtained using Gaussian sketching (GS) or Newton-Schulz iterations (NS). We report the relative Frobenius error $\|\widehat{S}(M_N) - \mathrm{msgn}(M)\|_F / \|\mathrm{msgn}(M)\|_F$ and the alignment error $1 - \langle \widehat{S}(M_N), \mathrm{msgn}(M) \rangle / (\|\widehat{S}(M_N)\|_F \|\mathrm{msgn}(M)\|_F)$.

Both methods are implemented in the same way as in the above synthetic experiments to illustrate the *computational efficiency*.

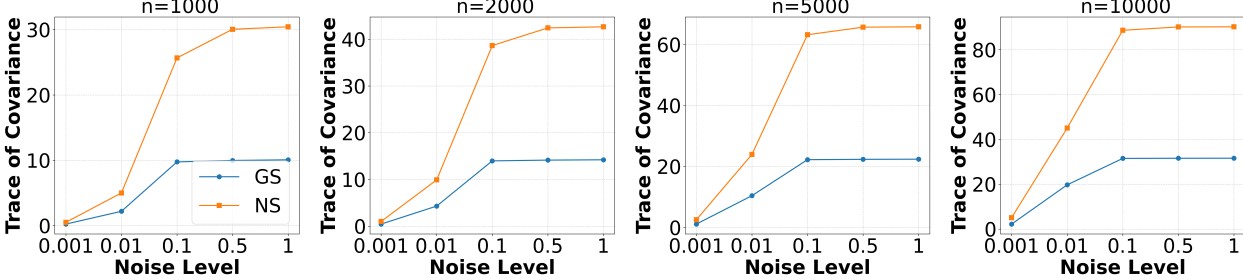

Figure 3: Comparison of variance levels in matrix sign estimation across Newton-Schulz iterations (NS), and our low-rank orthogonalization methods using Gaussian sketching (GS), applied to nearly low-rank matrices in the presence of noise.

Figure 3 presents a comparison of the average trace of the covariance matrices across all competing methods for matrix sign estimation. Figure 4 compares the estimated matrix sign operators with the ground-truth matrix sign operator. From these figures, we observe that, compared to applying the Newton-Schulz iterations to the full matrix, our low-rank orthogonalization method yields matrix sign estimates that are significantly less sensitive to noise. This is because singular vectors associated with small singular values are more sensitive to noise than others, so removing them improves the robustness of the matrix sign computation. As a result, our low-rank method provides more stable and robust estimates of the matrix sign function, particularly for nearly low-rank matrices.

## 3.2   Low-Rank Matrix-Signed Gradient Descent

In this subsection, we propose low-rank MSGD methods, including a fixed-rank variant and a safeguarded variant with adaptive ranks. Both variants use low-rank orthogonalization as a subroutine.

At each iteration $k \geq 0$, the fixed-rank variant invokes Algorithm 1 to compute $M_O^k = \mathrm{msgn}(M_Q^k)$, where $M_Q^k$ is a low-rank approximation of $\nabla f(X^k)$. Then, the next iterate $X^{k+1}$ is obtained by performing a line search update from $X^k$ along the matrix-signed direction $-M_O^k$. Details of this method are presented in Algorithm 2.

---

[1]For each $M$, we first calculate a set of estimates of $\mathrm{msgn}(M)$ in the presence of noise, denoted by $\{\mathrm{msgn}(M_{N_i})\}$, where $\{N_i\}$ denotes the set of stimulated noise. We vectorize these estimates and compute their empirical covariance matrix, and then use the trace of this covariance matrix to quantify the overall variability of the estimates.

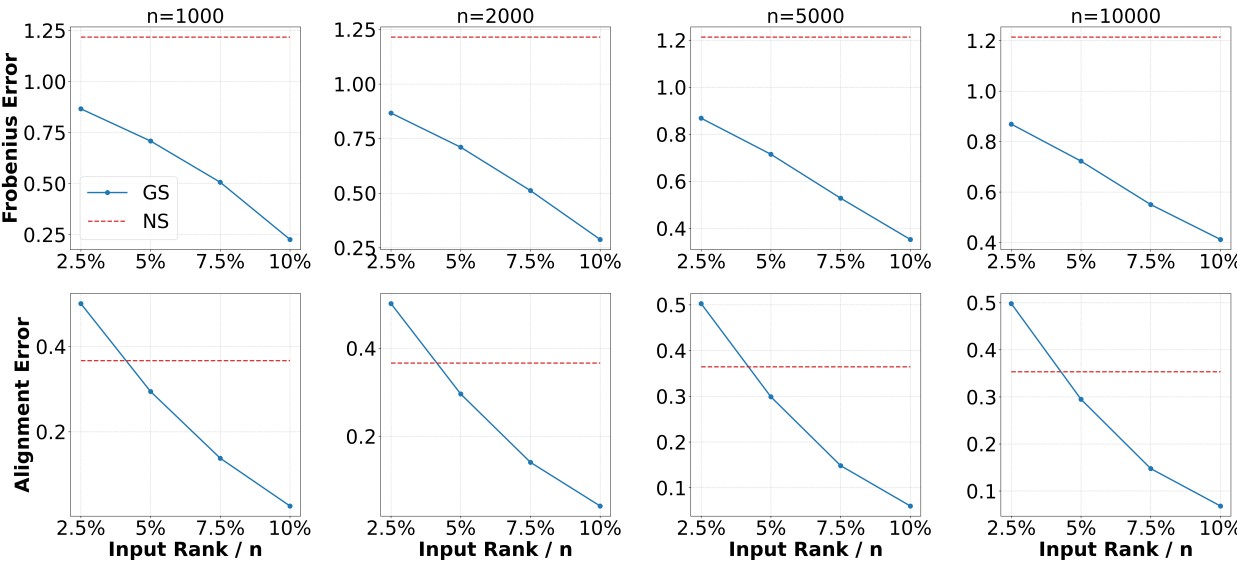

Figure 4: Comparison between the estimated and ground-truth matrix sign operators using Newton-Schulz iterations (NS) and our low-rank orthogonalization method based on Gaussian sketching (GS), applied to nearly low-rank matrices in the presence of noise.

---

**Algorithm 2** Low-Rank MSGD

**Input:** starting point $X^0 \in \mathbb{R}^{m \times n}$, rank parameter $r \in \mathbb{Z}_+ \cap [1, \varrho]$, step sizes $\{\eta_k\} \subset (0, \infty)$.
**for** $k = 0, 1, 2, \ldots$ **do**

Call Algorithm 1 (see Section 3.1) with $(M, r) = (\nabla f(X^k), r)$ to obtain an approximate matrix sign $M_O^k$, such that $M_O^k = \mathrm{msgn}(M_Q^k)$, where $M_Q^k$ is a low-rank approximation for $\nabla f(X^k)$.

Update the next iterate: $X^{k+1} = X^k - \eta_k M_O^k$.
**end for**

---

The following theorem provides a convergence guarantee for Algorithm 2, whose proof is deferred to Section B.2.

**Theorem 2.** *Suppose that Assumption 1 holds. Let $\{(X^k, M_O^k)\}$ be the sequence generated by Algorithm 2 with $M_O^k = \mathrm{msgn}(M_Q^k)$ for all $k \geq 0$ and step sizes $\{\eta_k\}$ given by $\eta_k = (k+1)^{-1/2}$ for all $k \geq 0$. Then, it holds that for all $K \geq 3$,*

$$\min_{0 \leq k \leq K-1} \{\|\nabla f(X^k)\|\} \leq \frac{f(X^0) - f_{\mathrm{low}} + L_* \ln K}{K^{1/2}} + \frac{2}{K^{1/2}} \sum_{k=0}^{K-1} \frac{\|\nabla f(X^k) - M_Q^k\|_*}{(k+1)^{1/2}}, \tag{6}$$

*where $L_*$ is defined in Assumption 1.*

**Remark 2.** *From Theorem 2, we observe that when $\{(X^k, M_O^k)\}$ satisfies $\sum_{k=0}^{K-1} \|\nabla f(X^k) - M_Q^k\|_*(k+1)^{-1/2} = \widetilde{\mathcal{O}}(1)$, it holds that $\min_{0 \leq k \leq K-1}\{\|\nabla f(X^k)\|_*\} = \widetilde{\mathcal{O}}(K^{-1/2})$, which matches, up to a logarithmic factor, the well-established optimal convergence rate for nonconvex optimization (e.g., see (Carmon et al., 2020)). Moreover, if the gradients $\{\nabla f(X^k)\}$ have low rank when $k$ is sufficiently large (as is often the case during deep neural network training (Zhao et al., 2024)), we can tune the rank parameter $r$ to be close to the effective rank of the gradients, causing the sequence $\{\|\nabla f(X^k) - M_Q^k\|_*\}$ to remain close to zero.*

We next describe a safeguarded low-rank MSGD method with adaptively updated ranks. At each iteration $k \geq 0$, this method invokes Algorithm 1 with $(M, r) = (\nabla f(X^k), r_k)$ to obtain $M_O^k = \mathrm{msgn}(M_Q^k)$, where $M_Q^k$ is a low-rank approximation of $\nabla f(X^k)$ such that the approximation error satisfies (7). Then, the next iterate

$X^{k+1}$ is obtained by performing a line search update from $X^k$ along the matrix-signed direction $-M_O^k$, with a suitable step size. Details of this method are provided in Algorithm 3.

It is noteworthy that the approximation error in (7) holds for some $r_k \leq \varrho$ because when $r_k = \varrho$, the error $\|M_Q^k - \nabla f(X^k)\|_*$ is zero. When the matrix is low-rank or has a small effective rank, $r_k$ can be chosen to be much smaller than $\varrho$. In practice, one can gradually increase the trial ranks to find $r_k$ such that (7) holds.

---

**Algorithm 3** Safeguarded Low-Rank MSGD

---

**Input:** starting point $X^0 \in \mathbb{R}^{m \times n}$, initial rank trial $r_0 \in \mathbb{Z}_+ \cap [1, \varrho]$, step sizes $\{\eta_k\} \subset (0, \infty)$, control errors $\{\delta_k\} \subset (0, \infty)$
**for** $k = 0, 1, 2, \ldots$ **do**
    Call Algorithm 1 (see Section 3.1) with $(M, r) = (\nabla f(X^k), r_k)$ to obtain an approximate matrix sign $M_O^k$ such that $M_O^k = \mathrm{msgn}(M_Q^k)$ and

$$\|\nabla f(X^k) - M_Q^k\|_* \leq \delta_k. \tag{7}$$

    Update the next iterate: $X^{k+1} = X^k - \eta_k M_O^k$.
**end for**

---

The following theorem establishes an iteration complexity of Algorithm 3, whose proof is deferred to Section B.2.

**Theorem 3.** *Suppose that Assumption 1 holds. Let $f_{\mathrm{low}}$ and $L_*$ be given in Assumption 1, and define*

$$U_{\mathrm{gd}} := f(X^0) - f_{\mathrm{low}} + L_* + 4. \tag{8}$$

*Let $\{X^k\}$ be generated by Algorithm 3 with inputs $\{(\eta_k, \delta_k)\}$ given by $\eta_k = \delta_k = (k+1)^{-1/2}$ for all $k \geq 0$. Then, for any $\epsilon \in (0, 1)$, it holds that $\min_{0 \leq k \leq K-1}\{\|\nabla f(X^k)\|_*\} \leq \epsilon$ for all $K$ satisfying*

$$K \geq \max\left\{\left(\frac{4U_{\mathrm{gd}}}{\epsilon}\ln\left(\frac{4U_{\mathrm{gd}}}{\epsilon}\right)\right)^2, 3\right\}.$$

**Remark 3.** *From Theorem 3, we observe that Algorithm 3 achieves an iteration complexity of $\widetilde{\mathcal{O}}(\epsilon^{-2})$ for finding an $\epsilon$-NSP of (1). This complexity bound matches, up to a polylogarithmic factor, the lower complexity bound as established in (Carmon et al., 2020).*

### 3.3 Low-Rank Muon

In this subsection, we propose a low-rank variant of Muon (Jordan et al., 2024b), and analyze its iteration complexity under heavy-tailed noise.

This method follows a framework similar to Muon (Jordan et al., 2024b), but instead of directly computing the matrix sign of the momentum update, it computes only its low-rank approximation. Specifically, at each iteration of this method, it first performs a momentum update to generate $M^k$ by aggregating stochastic gradients of $f$ evaluated at $X^0, \ldots, X^k$. Then, Algorithm 1 is invoked to obtain $M_O^k = \mathrm{msgn}(M_Q^k)$, where $M_Q^k$ is a low-rank approximation of $\nabla f(X^k)$ such that the approximation error satisfies (10). The next iterate $X^{k+1}$ is obtained by performing a line search update from $X^k$ along the matrix-signed direction $-M_O^k$ with a suitable step size. Details of this method are given in Algorithm 4.

Before analyzing the complexity of Algorithm 4 for computing an approximate solution to (1), we make the following heavy-tailed noise assumption regarding the stochastic gradient $G(\cdot; \xi)$.

**Assumption 2.** *The stochastic gradient estimator $G : \mathbb{R}^{m \times n} \times \Xi \to \mathbb{R}^{m \times n}$ satisfies*

$$\mathbb{E}[G(X; \xi)] = \nabla f(X), \quad \mathbb{E}[\|G(X; \xi) - \nabla f(X)\|_F^\alpha] \leq \sigma^\alpha$$

*for some $\sigma > 0$ and $\alpha \in (1, 2]$.*

---

**Algorithm 4** Low-Rank Muon

---

**Input:** starting point $X^0 \in \mathbb{R}^{m \times n}$, initial rank trial $r_0 \in \mathbb{Z}_+ \cap [1, \varrho]$, step sizes $\{\eta_k\} \subset (0, \infty)$, weighting parameters $\{\theta_k\} \subset (0, 1]$, control errors $\{\delta_k\} \subset (0, \infty)$.
**Initialize:** $M^{-1} = \mathbf{0}_{m \times n}$ and $\theta_{-1} = 1$.
**for** $k = 0, 1, 2, \ldots$ **do**
    Compute the full-rank search direction:

$$M^k = (1 - \theta_{k-1})M^{k-1} + \theta_{k-1}G(X^k; \xi^k). \tag{9}$$

    Call Algorithm 1 (see Section 3.1) with $(M, r) = (M^k, r_k)$ to obtain an approximate matrix sign $M_O^k$ such that $M_O^k = \mathrm{msgn}(M_Q^k)$ and

$$\|M^k - M_Q^k\|_* \leq \delta_k. \tag{10}$$

    Update the next iterate:

$$X^{k+1} = X^k - \eta_k M_O^k. \tag{11}$$

**end for**

---

The following theorem establishes the iteration complexity of Algorithm 4, whose proof is deferred to Section B.3.

**Theorem 4.** *Suppose that Assumptions 1 and 2 hold. Let $f_{\mathrm{low}}$ and $L_*$ be given in Assumption 1, and $\sigma$ and $\alpha$ be given in Assumption 2. Define*

$$U_{\mathrm{mn}} := f(X^0) - f_{\mathrm{low}} + \sigma^\alpha + 2L_* + 4 + 2(\alpha - 1)(2\varrho^{1/2}/\alpha)^{\alpha/(\alpha-1)} + 6L_*^\alpha + 4\sigma^\alpha. \tag{12}$$

*Let $\{X^k\}$ be generated by Algorithm 4 with input parameters $\{(\eta_k, \theta_k, \delta_k)\}$ given by $\eta_k = (k+1)^{-(2\alpha-1)/(3\alpha-2)}$, $\theta_k = (k+1)^{-\alpha/(3\alpha-2)}$, and $\delta_k = (k+1)^{-(\alpha-1)/(3\alpha-2)}$ for all $k \geq 0$. Then, for any $\epsilon \in (0, 1)$, it holds that $\mathbb{E}[\|\nabla f(X^{\iota_K})\|_*] \leq \epsilon$ for all $K$ satisfying*

$$K \geq \max \left\{ \left( \frac{2(3\alpha - 2)U_{\mathrm{mn}}}{(\alpha - 1)\epsilon} \ln \left( \frac{2(3\alpha - 2)U_{\mathrm{mn}}}{(\alpha - 1)\epsilon} \right) \right)^{\frac{3\alpha-2}{\alpha-1}}, 3 \right\},$$

*where $\iota_K$ is uniformly drawn from $\{0, \ldots, K - 1\}$.*

**Remark 4.** *(i) From Theorem 4, one can observe that Algorithm 4 achieves an iteration complexity of $\widetilde{\mathcal{O}}(\epsilon^{-(3\alpha-2)/(\alpha-1)})$ for finding an $\epsilon$-SNSP of problem (1), which matches the optimal dependence on $\epsilon$ in the complexity results for vector-variate stochastic first-order methods under heavy-tailed noise (see, e.g., (He et al., 2025a; Zhang et al., 2020)). To the best of our knowledge, our result is the first to show that the inexact Muon-type algorithms can achieve an iteration complexity with optimal dependence on $\epsilon$ when applied to matrix-variate optimization under heavy-tailed noise.*

*(ii) Notice that, to ensure the optimal stationarity guarantee for the general matrix optimization problem (1), Theorem 4 requires the tolerance sequence $\{\delta_k\}$ to satisfy $\delta_k = \mathcal{O}(k^{-(\alpha-1)/(3\alpha-2)})$, which matches the convergence rate of the stationarity measure up to a polylogarithmic factor. However, to ensure convergence alone, it suffices that $\{\delta_k\}$ decay at an arbitrary polynomial rate. If it decays more slowly than the rate specified in Theorem 4, the stationarity measure converges at the rate of $\{\delta_k\}$, up to a polylogarithmic factor.*

*(iii) For GPT-2 and LLaMA pretraining, the condition that $\{\delta_k\}$ decays at the rate of $\mathcal{O}(k^{-(\alpha-1)/(3\alpha-2)})$ is not restrictive because of the benign low-rank structure of the gradients. In Figures 5 and 8, we numerically verify that $\{\delta_k\}$ is dominated by $\{(k+1)^{-1/2}\}$ by plotting the estimation errors for the gradients associated with the Q, K, and V matrices in a particular block during pretraining.*

## 4 Numerical Experiments

In this section, we present numerical experiments to evaluate the performance of low-rank Muon (Algorithm 4) and compare our method with vanilla Muon, AdamW, and SGD. The experiments are conducted on GPT-2 pretraining (Section 4.1) and LLaMA pretraining (Section 4.2). All experiments are executed on a server with two NVIDIA A100 GPUs (80 GB). The code to implement the proposed algorithm on these numerical examples is available at `https://anonymous.4open.science/r/Low-rank-Muon-75C6/`.

We now describe a heuristic that adaptively updates the ranks $\{r_k\}$ in Algorithm 4. At each iteration $k \geq 1$, we use the stable rank of $M^{k-1}$, i.e., $\|M^{k-1}\|_F^2 / \|M^{k-1}\|^2$, as a numerical proxy for its rank and restrict the resulting estimate to $[64, 256]$ in increments of 32. In our implementation, we use power iteration to estimate $\|M^{k-1}\|$ and set $r_0 = 64$.

### 4.1 GPT-2 Pretraining

In this subsection, we consider pretraining GPT-2 (Radford et al., 2019), a transformer-based language model. We experiment with GPT-2 models of sizes 60M, 135M, 350M, and 1B on the same three datasets tested in the Muon GitHub repository (Jordan et al., 2024b): FineWeb10B, FineWeb100B, and FineWebEdu10B.

We apply low-rank Muon, Muon, SOAP, GaLore, AdamW, and SGD with momentum (SGDM) for GPT-2 pretraining. We implement two low-rank Muon variants with fixed ranks of 100 and 200, abbreviated as LR-Muon100 and LR-Muon200, respectively, as well as an adaptive-rank variant, abbreviated as AdaRank. Following the experiments in (Jordan et al., 2024a; Lau et al., 2025), we use AdamW to train the embedding and head layers for both Muon and low-rank Muon. We randomly initialize all methods using the same seed, without relying on pretrained checkpoints, and terminate all methods after one training epoch, consisting of 5960 iterations. We compare all methods using validation perplexity, a standard metric in foundation model training (e.g., (Radford et al., 2019; Touvron et al., 2023; Vaswani et al., 2017)), which is defined as the exponential of the validation loss and serves to amplify performance differences. The algorithmic parameters are carefully tuned to suit each method well in terms of computational performance. More details about the experimental setups, including algorithm and GPT-2 model parameters, are provided in Appendix A.

In Figure 5, we present the sequence of estimation errors, referred to as the nuclear tail, $\{\|M^k - M_Q^k\|_*\}$, during the first 5,000 iterations of GPT-2 pretraining using low-rank Muon with different rank choices. We observe that the nuclear tail is dominated by $\{(k+1)^{-1/2}\}$ and decreases slightly as the input rank increases. Therefore, the condition on $\{\delta_k\}$ in Theorem 4 is satisfied. In addition, we visualize the top 100 singular values of the momentum updates $\{M^k\}$ associated with the Q, K, and V matrices of a 350M-parameter GPT-2 model trained using Muon on the FineWeb100B dataset in Figure 6. The figure shows these singular values across different training iterations, providing partial evidence for the low-rank structure of the momentum updates.

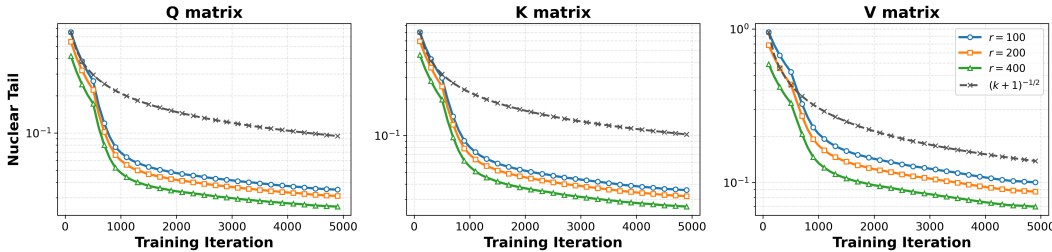

Figure 5: Estimation errors, i.e., $\{\|M^k - M_Q^k\|_*\}$, associated with the Q, K, and V matrices during the first 5.960 optimizer iterations of a GPT-2 (1B) pretraining run. The colored curves show the results for low-rank Muon with fixed ranks of 100, 200, and 400. The gray dashed curve denotes the reference sequence $\{(k+1)^{-1/2}\}$, which represents the most stringent condition in Theorems 3 and 4.

We present a comparison of validation perplexity and computational time in Table 1. From this table, we observe that for GPT-2 with a model size of 60M, our low-rank Muon achieves slightly worse validation

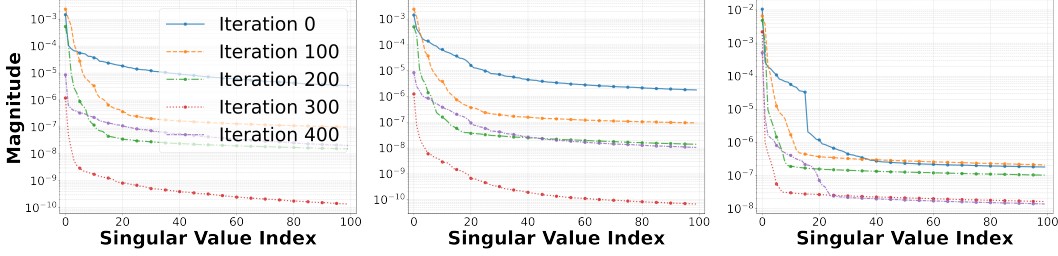

Figure 6: Singular value distributions of the Q, K, and V matrices over training iterations at layer 16 during GPT-2 (350M) pretraining.

Table 1: Comparison of validation perplexity and computational time for all competing methods on GPT-2 pretraining. The ± values report empirical variability over three runs with different random seeds.

| Dataset | Method | Validation Perplexity | | | | Computational Time | | | |
|---|---|---|---|---|---|---|---|---|---|
| | | 60M | 135M | 350M | 1B | 60M | 135M | 350M | 1B |
| FineWeb10B | SGDM | 48.85±0.44 | 77.50±0.57 | 31.36±0.33 | 27.86±0.29 | 2.51e3 | 7.08e3 | 1.20e4 | 2.70e5 |
| | AdamW | 43.56±0.35 | 43.56±0.32 | 25.72±0.27 | 21.58±0.23 | 2.52e3 | 7.08e3 | 1.21e4 | 2.69e5 |
| | GaLore | 57.61±0.49 | 39.59±0.37 | 30.87±0.31 | 28.58±0.28 | 2.73e3 | 7.12e3 | 1.24e4 | 2.76e5 |
| | SOAP | 41.32±0.38 | 55.37±0.46 | 38.45±0.34 | 34.58±0.30 | 2.56e3 | 7.09e3 | 1.23e4 | 2.73e5 |
| | Muon | **32.89**±0.27 | 32.99±0.26 | 28.48±0.24 | 20.99±0.21 | 2.76e3 | 7.57e3 | 1.32e4 | 3.04e5 |
| | LR-Muon100 | 35.21±0.30 | 28.83±0.25 | 27.32±0.23 | 21.70±0.22 | 2.69e3 | 7.49e3 | 1.25e4 | 2.90e5 |
| | LR-Muon200 | 33.98±0.28 | **27.34**±0.24 | **25.64**±0.22 | **20.69**±0.20 | 2.79e3 | 7.54e3 | 1.26e4 | 2.91e5 |
| | AdaRank | 35.61±0.31 | 28.31±0.27 | 26.75±0.24 | 21.45±0.22 | 2.72e3 | 7.55e3 | 1.25e4 | 2.91e5 |
| FineWeb100B | SGDM | 51.48±0.47 | 51.48±0.43 | 31.54±0.32 | 29.75±0.31 | 2.69e3 | 7.25e3 | 1.04e4 | 2.52e5 |
| | AdamW | 43.52±0.36 | 43.51±0.33 | 27.47±0.28 | 23.53±0.24 | 2.70e3 | 7.21e3 | 1.05e4 | 2.56e5 |
| | GaLore | 56.42±0.48 | 58.09±0.50 | 31.19±0.32 | 26.53±0.27 | 2.81e3 | 7.31e3 | 1.15e4 | 2.67e5 |
| | SOAP | 41.32±0.37 | 44.32±0.39 | 30.98±0.29 | 25.56±0.25 | 2.71e3 | 7.23e3 | 1.07e4 | 2.58e5 |
| | Muon | 34.04±0.29 | 33.02±0.27 | 32.76±0.26 | 20.75±0.20 | 2.79e3 | 7.70e3 | 1.19e4 | 3.09e5 |
| | LR-Muon100 | 35.78±0.31 | 30.19±0.26 | 27.59±0.24 | 21.76±0.21 | 2.77e3 | 7.62e3 | 1.09e4 | 2.87e5 |
| | LR-Muon200 | **34.02**±0.29 | **28.31**±0.25 | **25.86**±0.23 | **20.45**±0.19 | 2.80e3 | 7.69e3 | 1.10e4 | 2.91e5 |
| | AdaRank | 35.93±0.32 | 29.58±0.28 | 26.76±0.30 | 21.02±0.23 | 2.79e3 | 7.68e3 | 1.08e4 | 2.89e5 |
| FineWebEdu10B | SGDM | 50.92±0.45 | 32.79±0.33 | 31.54±0.32 | 25.49±0.26 | 2.56e3 | 7.09e3 | 1.20e4 | 2.53e5 |
| | AdamW | 43.56±0.34 | 26.45±0.27 | 21.01±0.22 | 22.45±0.24 | 2.59e3 | 7.09e3 | 1.20e4 | 2.56e5 |
| | GaLore | 69.49±0.53 | 39.59±0.36 | 57.42±0.47 | 37.45±0.35 | 2.64e3 | 7.12e3 | 1.24e4 | 2.60e5 |
| | SOAP | 55.37±0.45 | 41.77±0.38 | 47.90±0.41 | 39.20±0.36 | 2.60e3 | 7.11e3 | 1.22e4 | 2.58e5 |
| | Muon | 32.88±0.28 | 23.89±0.23 | 22.23±0.22 | 19.54±0.19 | 2.80e3 | 7.63e3 | 1.33e4 | 3.05e5 |
| | LR-Muon100 | 35.60±0.30 | 23.50±0.22 | 22.39±0.21 | 19.97±0.20 | 2.74e3 | 7.53e3 | 1.26e4 | 2.85e5 |
| | LR-Muon200 | 33.93±0.27 | **22.37**±0.21 | **20.96**±0.20 | **18.85**±0.18 | 2.83e3 | 7.59e3 | 1.29e4 | 2.87e5 |
| | AdaRank | **32.51**±0.26 | 23.20±0.23 | 21.20±0.22 | 18.94±0.21 | 2.73e3 | 7.58e3 | 1.26e4 | 2.84e5 |

perplexity than full-rank Muon, but performs better than AdamW, SGDM, SOAP, and GaLore. For GPT-2 models with larger sizes, our low-rank Muon achieves better validation perplexity than all other competing methods. We also observe that the computational time of our low-rank Muon improves upon that of vanilla Muon, although all Muon-type methods remain slower than SGDM and AdamW. These observations show that our low-rank orthogonalization enables obtaining more generalizable GPT-2 models when the model size is large, but may be less effective for smaller models. In addition, our low-rank orthogonalization saves computational time compared to orthogonalization by Newton-Schulz iterations, but the gain is not as significant as those presented in Section 3.1, since the forward and backward passes of neural networks account for the majority of the computational time. In addition, we observe that the adaptive-rank variant of low-rank Muon achieves performance comparable to that of the fixed-rank variants with ranks 100 and 200, while alleviating the effort required for rank-specific hyperparameter tuning.

We plot the convergence behavior of validation perplexity versus training iterations in Figure 7. From this figure, we observe that for each model size, the competing methods exhibit similar performance patterns across the three datasets. For the model size of 60M, all Muon-type methods outperform AdamW and SGDM, while our low-rank Muon performs slightly worse than vanilla Muon. For the 135M model, our low-rank

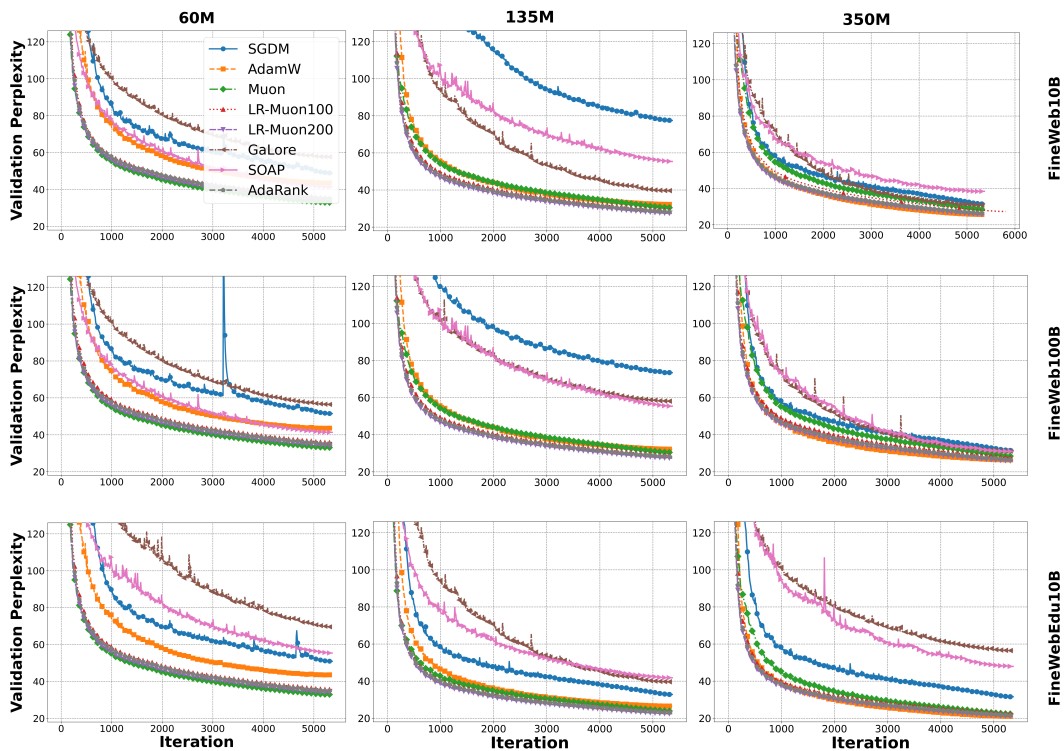

Figure 7: Comparison of validation perplexity versus computational time for all competing methods in training GPT-2 models.

Muon outperforms vanilla Muon and other training methods. Vanilla Muon performs similarly to AdamW, and all methods show a large improvement compared to SGDM. For the model size of 350M, our low-rank Muon has slightly better performance than AdamW, while outperforming Muon and SGDM by a larger margin. These observations support that our low-rank Muon generally improves upon vanilla Muon in GPT-2 training and is more effective for models with relatively larger sizes.

## 4.2 LLaMA Pretraining

In this subsection, we consider pretraining LLaMA (Touvron et al., 2023), a transformer-based language model with a more refined architecture than GPT-2. We experiment with LLaMA models of sizes 60M, 135M, 350M, and 1B on the same three datasets tested in the Muon GitHub repository (Jordan et al., 2024b): FineWeb10B, FineWeb100B, and FineWebEdu10B.

We apply low-rank Muon, Muon, AdamW, GaLore, SOAP and SGD with momentum (SGDM) for pretraining LLaMA. Similar to Section 4.1, we implement two versions of low-rank Muon with fixed ranks of 100 and 200, as well as an adaptive-rank variant of low-rank Muon, and we use AdamW to train the embedding and head layers for both Muon and low-rank Muon. We randomly initialize all methods using the same seed, without relying on pretrained checkpoints, and terminate all methods after one training epoch, consisting of 5.960 iterations. We compare all methods using validation perplexity. The algorithmic parameters are carefully tuned to suit each method well in terms of computational performance. More details about the experimental setups, including algorithm and LLaMA model parameters, are provided in Appendix A.

In Figure 8, we present the sequence of estimation errors, referred to as the nuclear tail, $\{\|M^k - M_Q^k\|_*\}$, during the first 5.960 iterations of LLaMA pretraining using low-rank Muon with different rank choices. We observe that the nuclear tail is dominated by $\{(k+1)^{-1/2}\}$ and decreases slightly as the input rank increases. Therefore, the condition on $\{\delta_k\}$ in Theorem 4 is satisfied. In addition, we visualize the top 100 singular values of the momentum updates $\{M^k\}$ associated with the Q, K, and V matrices of a 350M-parameter LLaMA

model trained using Muon on the FineWeb100B dataset in Figure 9. The figure shows these singular values across different training iterations, providing partial evidence for the low-rank structure of the momentum updates.

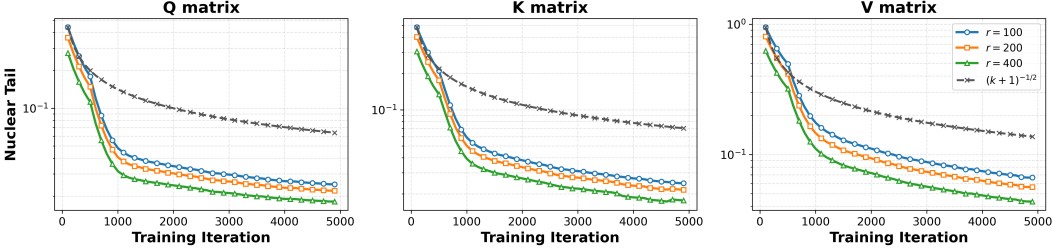

Figure 8: Estimation errors, i.e., $\{\|M^k - M_Q^k\|_*\}$, associated with the Q, K, and V matrices during the first 5,000 optimizer iterations of a LLaMA (1B) pretraining run. The colored curves show the results for low-rank Muon with fixed ranks of 100, 200, and 400. The gray dashed curve denotes the reference sequence $\{(k+1)^{-1/2}\}$, which represents the most stringent condition in Theorems 3 and 4.

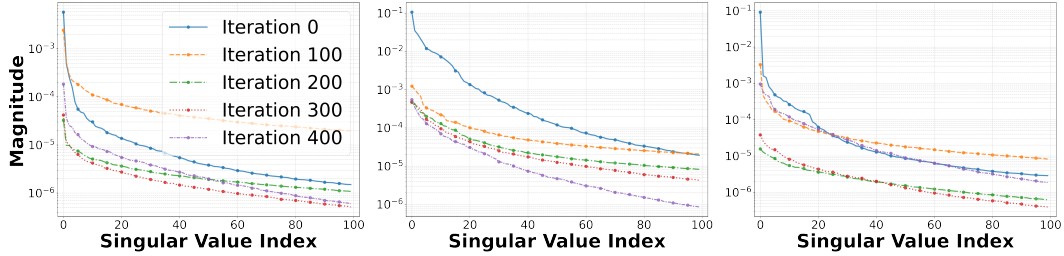

Figure 9: Singular value distributions of the Q, K, and V matrices over training iterations at layer 16 during LLaMA (350M) pretraining.

We present a comparison of validation perplexity and computational time in Table 2. From this table, we observe that our low-rank Muon consistently achieves better validation perplexity compared to Muon, GaLore, and SOAP for the majority of model sizes and tested datasets, while vastly improving upon the validation perplexity achieved by AdamW and SGDM. We also observe that the computational time of our low-rank Muon improves upon that of vanilla Muon, though all Muon-type methods remain slower than SGDM and AdamW. These observations demonstrate that our low-rank orthogonalization produces more generalizable LLaMA models across all tested model sizes and datasets. Moreover, it reduces computational time compared to the Newton-Schulz iterations for orthogonalization, although the improvement is less significant than the gains discussed in Section 3.1, as most of the computational time is dominated by the forward and backward passes of the neural networks. In addition, we observe that the adaptive-rank variant of low-rank Muon achieves performance comparable to that of the fixed-rank versions with ranks of 100 and 200, while reducing the effort required for hyperparameter tuning.

Table 2: Comparison of validation perplexity and computational time for all competing methods on LLaMA pretraining. The ± values report empirical variability over three runs with different random seeds.

| Dataset | Method | Validation Perplexity | | | | Computational Time | | | |
|---|---|---|---|---|---|---|---|---|---|
| | | 60M | 135M | 350M | 1B | 60M | 135M | 350M | 1B |
| FineWeb10B | SGDM | 94.90±0.64 | 85.99±0.59 | 86.49±0.58 | 44.71±0.38 | 2.49e3 | 5.27e3 | 1.20e4 | 2.77e5 |
| | AdamW | 50.51±0.42 | 40.17±0.35 | 35.76±0.31 | 24.14±0.25 | 2.46e3 | 5.27e3 | 1.21e4 | 2.78e5 |
| | GaLore | 53.69±0.68 | 47.70±0.51 | 38.12±0.52 | 26.54±0.30 | 2.37e3 | 6.58e3 | 1.18e4 | 2.75e5 |
| | SOAP | 49.23±0.45 | 50.23±0.35 | 41.73±1.08 | 26.34±0.28 | 2.40e3 | 5.67e3 | 1.11e4 | 2.67e5 |
| | Muon | 36.68±0.31 | 37.96±0.32 | 36.43±0.30 | 20.99±0.21 | 2.59e3 | 5.69e3 | 1.35e4 | 3.18e5 |
| | LR-Muon100 | 37.33±0.32 | **34.08**±0.28 | 33.54±0.27 | 21.70±0.22 | 2.69e3 | 5.66e3 | 1.28e4 | 2.79e5 |
| | LR-Muon200 | **36.05**±0.30 | 34.37±0.29 | **32.37**±0.27 | **20.69**±0.20 | 2.70e3 | 5.70e3 | 1.30e4 | 2.91e5 |
| | AdaRank | 38.15±0.37 | 34.96±0.34 | 33.80±0.30 | 22.98±0.22 | 2.69e3 | 5.69e3 | 1.29e4 | 2.90e5 |
| FineWeb100B | SGDM | 81.59±0.55 | 77.50±0.52 | 87.49±0.60 | 37.56±0.34 | 2.60e3 | 5.44e3 | 1.05e4 | 2.65e5 |
| | AdamW | 41.49±0.36 | 40.54±0.34 | 35.47±0.30 | 23.89±0.24 | 2.61e3 | 5.47e3 | 1.06e4 | 2.69e5 |
| | GaLore | 58.97±0.46 | 45.14±0.38 | 37.05±0.33 | 35.56±0.31 | 2.65e3 | 5.50e3 | 1.09e4 | 2.69e5 |
| | SOAP | 55.08±0.44 | 52.28±0.42 | 47.52±0.39 | 43.89±0.36 | 2.64e3 | 5.50e3 | 1.07e4 | 2.70e5 |
| | Muon | 38.49±0.33 | 37.66±0.31 | 36.66±0.30 | 21.76±0.22 | 2.58e3 | 5.63e3 | 1.07e4 | 3.16e5 |
| | LR-Muon100 | 37.68±0.32 | 34.58±0.29 | 33.71±0.28 | 22.86±0.23 | 2.63e3 | 5.55e3 | 1.10e4 | 2.79e5 |
| | LR-Muon200 | 36.26±0.31 | **34.20**±0.28 | **32.73**±0.27 | **20.98**±0.21 | 2.75e3 | 5.69e3 | 1.11e4 | 2.88e5 |
| | AdaRank | **34.90**±0.36 | 34.56±0.33 | 32.89±0.45 | 21.89±0.30 | 2.77e3 | 5.78e3 | 1.11e4 | 2.82e5 |
| FineWebEdu10B | SGDM | 82.63±0.56 | 73.48±0.50 | 73.48±0.49 | 47.56±0.39 | 2.57e3 | 5.27e3 | 1.20e4 | 2.76e5 |
| | AdamW | 41.82±0.35 | 32.96±0.29 | 29.12±0.26 | 22.54±0.23 | 2.60e3 | 5.29e3 | 1.20e4 | 2.78e5 |
| | GaLore | 64.04±0.49 | 68.13±0.52 | 45.13±0.37 | 43.56±0.35 | 2.57e3 | 5.27e3 | 1.20e4 | 2.76e5 |
| | SOAP | 40.32±0.34 | 28.10±0.27 | 27.79±0.26 | 24.54±0.24 | 2.60e3 | 5.29e3 | 1.20e4 | 2.78e5 |
| | Muon | **28.57**±0.25 | 30.72±0.27 | 29.47±0.26 | 22.67±0.22 | 2.68e3 | 5.84e3 | 1.33e4 | 3.25e5 |
| | LR-Muon100 | 30.52±0.27 | 27.72±0.25 | 27.24±0.24 | 21.36±0.21 | 2.72e3 | 5.66e3 | 1.24e4 | 2.88e5 |
| | LR-Muon200 | 29.29±0.26 | **27.65**±0.24 | **26.87**±0.23 | **19.54**±0.19 | 2.81e3 | 5.70e3 | 1.26e4 | 2.98e5 |
| | AdaRank | 32.30±0.30 | 29.51±0.29 | 28.35±0.27 | 23.45±0.20 | 2.83e3 | 5.68e3 | 1.27e4 | 2.89e5 |

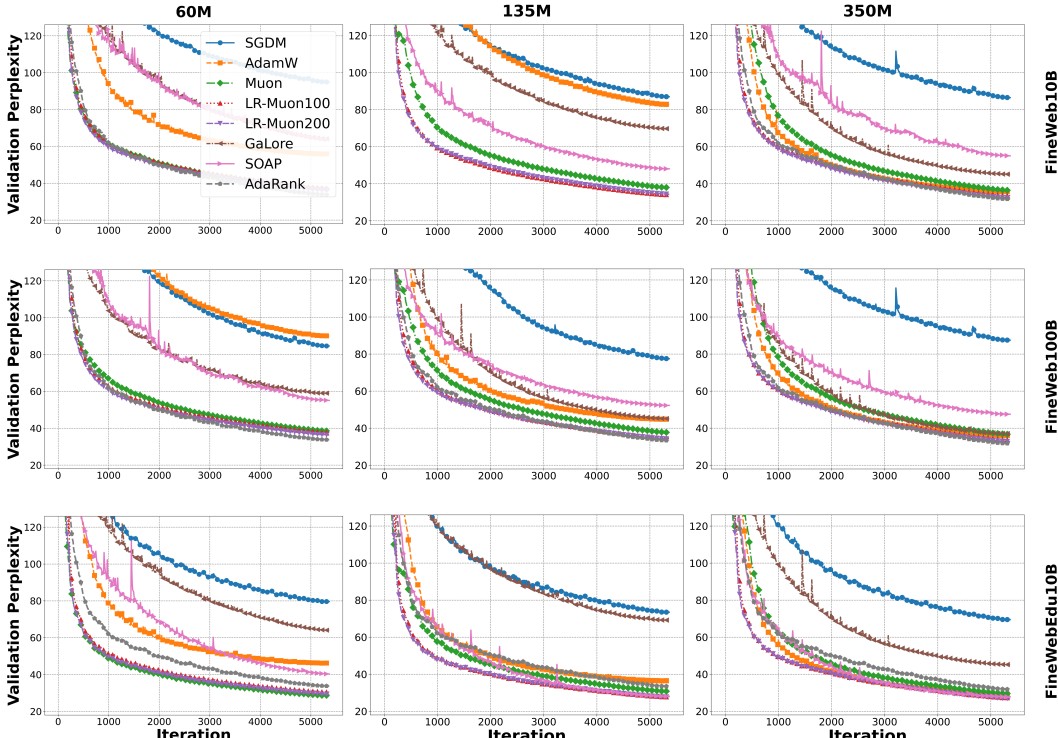

Figure 10: Comparison of validation perplexity and computational time for all competing methods in training LLaMA models.

We plot the convergence behavior of validation perplexity versus training iterations in Figure 10. From this figure, we observe that for the model size of 60M, all Muon-type methods outperforms AdamW and SGDM, while our low-rank Muon has similar performance compared to the vanilla Muon. For the 135M and 350M model size, our low-rank Muon outperforms the vanilla Muon and other training methods. These observations support that our low-rank Muon generally matching or improving upon vanilla Muon on LLaMA pretraining tasks, with clearer improvements observed for larger models.

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

# A    Experimental Details

In this part, we present the experimental settings and efficiency diagnostics for GPT-2 and LLaMA pretraining conducted in Section 4. We first introduce the model configurations, followed by an overall efficiency summary and detailed efficiency results for the completed adaptive-rank runs. Finally, we report the algorithmic hyperparameter search grids used for all methods.

## A.1    Model Configurations and Experimental Protocol

Table 3 presents the model configurations used in our GPT-2 and LLaMA experiments. All models are trained from random initialization without pretrained checkpoints. We conduct experiments on FineWeb10B, FineWeb100B, and FineWebEdu10B, with each run lasting 5,960 optimizer iterations. The batch size is 2, and the maximum training sequence length is 65,536 tokens. Thus, each iteration processes at most 131,072 tokens, and each complete run processes at most 655,360,000 tokens.

Validation is performed on a held-out set containing 10,485,760 tokens, with a maximum sequence length of 262,144 tokens. We report the validation perplexity, computed as the exponential of the validation loss. All experiments are conducted using two NVIDIA A100 GPUs with 80GB of memory each. In the main comparison tables, we report empirical variability over three runs with different random seeds.

Table 3: Hyperparameter configurations for GPT-2 and LLaMA models.

| Model Size | Hidden Dim. | FFN Dim. | # Heads | # Layers |
|:---:|:---:|:---:|:---:|:---:|
| 60M | 512 | 1376 | 8 | 8 |
| 135M | 768 | 2048 | 12 | 12 |
| 350M | 1024 | 2736 | 16 | 24 |
| 1B | 2048 | 5461 | 24 | 32 |

## A.2    Overall Efficiency Summary

Table 4 summarizes the computational efficiency of the competing methods using representative GPT-2 and LLaMA experiments. We report wall-clock time, throughput in tokens per second, time to reach a target perplexity, peak GPU memory, and optimizer overhead.

For a target perplexity $\tau$, the time to target is defined as the accumulated training time at the first validation checkpoint where the validation perplexity is less than $\tau$. Throughput is computed as the total number of processed training tokens divided by the wall-clock training time. Optimizer overhead is measured by timing the optimizer update separately from the forward and backward passes using CUDA events. For Muon-style methods, we report the overhead of matrix orthogonalization and parameter updates, excluding common optimizer bookkeeping. Peak memory is obtained from the final PyTorch CUDA peak-memory counters and includes both allocated and reserved memory.

## A.3    Completed-Run Efficiency

Table 5 provides detailed efficiency diagnostics for all completed adaptive-rank low-rank Muon runs. These results cover different model architectures, model sizes, and datasets. We report the final validation perplexity, wall-clock time, time to reach a perplexity of at most 50, throughput, and peak GPU memory.

Across these runs, the final validation perplexity ranges from 18.94 to 45.31, corresponding to validation losses between 2.94 and 3.81. Wall-clock time ranges from $2.69 \times 10^3$ to $2.91 \times 10^5$ seconds, while the time to reach a perplexity of at most 50 ranges from $1.13 \times 10^3$ to $2.33 \times 10^4$ seconds. Throughput ranges from $2.25 \times 10^3$ to $2.44 \times 10^5$ tokens per second, and peak allocated GPU memory ranges from 13.1GB to 76.3GB.

## A.4    Algorithmic Hyperparameters

Table 4: Computational efficiency results for GPT-2 and LLaMA pretraining. Training time is taken from Tables 1 and 2, and throughput is computed using a training budget of $5,960 \times 131,072$ tokens. Time to target denotes the time required to reach a validation perplexity of at most 50. Peak memory reports allocated and reserved GPU memory. For Muon-style methods, optimizer overhead refers to the matrix orthogonalization and parameter-update operations.

| Model | Method | Wall-clock (s) | Tokens/sec | Time to PPL $\leq 50$ (s) | Peak Mem. alloc./res. (MiB) | Optimizer overhead |
|---|---|---|---|---|---|---|
| GPT-2 | SGDM | $2.60 \times 10^4$ | $2.52 \times 10^4$ | $2.56 \times 10^4$ | 32922 / 33974 | 5% |
| GPT-2 | AdamW | $2.60 \times 10^4$ | $2.52 \times 10^4$ | $1.73 \times 10^4$ | 33084 / 34074 | 12% |
| GPT-2 | GaLore | $2.70 \times 10^4$ | $2.42 \times 10^4$ | $1.10 \times 10^4$ | 32828 / 33522 | 23% |
| GPT-2 | SOAP | $2.64 \times 10^4$ | $2.48 \times 10^4$ | $3.92 \times 10^4$ | 58676 / 65480 | 32% |
| GPT-2 | Muon | $2.92 \times 10^4$ | $2.24 \times 10^4$ | $3.05 \times 10^4$ | 33003 / 33774 | 42% |
| GPT-2 | LR-Muon100 | $2.79 \times 10^4$ | $2.35 \times 10^4$ | $1.41 \times 10^4$ | 33003 / 33494 | 16% |
| GPT-2 | LR-Muon200 | $2.85 \times 10^4$ | $2.30 \times 10^4$ | $1.18 \times 10^4$ | 33003 / 33714 | 20% |
| GPT-2 | AdaRank | $2.80 \times 10^4$ | $2.34 \times 10^4$ | $1.29 \times 10^4$ | 32317 / 32654 | 21% |
| LLaMA | SGDM | $2.63 \times 10^4$ | $2.50 \times 10^4$ | $3.07 \times 10^4$ | 22018 / 24754 | 5% |
| LLaMA | AdamW | $2.62 \times 10^4$ | $2.51 \times 10^4$ | $1.76 \times 10^4$ | 22144 / 24834 | 12% |
| LLaMA | GaLore | $2.56 \times 10^4$ | $2.56 \times 10^4$ | $1.99 \times 10^4$ | 32828 / 33522 | 23% |
| LLaMA | SOAP | $2.69 \times 10^4$ | $2.44 \times 10^4$ | $2.50 \times 10^4$ | 59806 / 61914 | 32% |
| LLaMA | Muon | $2.90 \times 10^4$ | $2.27 \times 10^4$ | $3.19 \times 10^4$ | 22081 / 23834 | 42% |
| LLaMA | LR-Muon100 | $2.75 \times 10^4$ | $2.38 \times 10^4$ | $1.64 \times 10^4$ | 22081 / 23774 | 15% |
| LLaMA | LR-Muon200 | $2.84 \times 10^4$ | $2.31 \times 10^4$ | $1.63 \times 10^4$ | 22081 / 23814 | 19% |
| LLaMA | AdaRank | $2.79 \times 10^4$ | $2.35 \times 10^4$ | $1.64 \times 10^4$ | 22270 / 23894 | 22% |

Table 5: Efficiency of AdaRank across GPT-2 and LLaMA pretraining configurations. Final validation perplexity and training time correspond to the AdaRank results in Tables 1 and 2. Time to target is measured at the first validation checkpoint reaching a perplexity of at most 50. Throughput is computed using the full training budget of $5,960 \times 131,072$ tokens. Peak GPU memory is reported as allocated/reserved memory in MiB.

| Model | Scale | Dataset | Final PPL | Wall-clock (s) | Time to PPL $\leq 50$ (s) | Tokens/sec | Peak Mem. (MiB) |
|---|---|---|---|---|---|---|---|
| GPT-2 | 60M | FineWeb10B | 35.61 | $2.72 \times 10^3$ | $1.52 \times 10^3$ | $2.41 \times 10^5$ | 14040 / 15712 |
| GPT-2 | 135M | FineWeb10B | 28.31 | $7.55 \times 10^3$ | $3.02 \times 10^3$ | $8.68 \times 10^4$ | 32317 / 32654 |
| GPT-2 | 350M | FineWeb10B | 26.75 | $1.25 \times 10^4$ | $3.75 \times 10^3$ | $5.24 \times 10^4$ | 48063 / 50362 |
| GPT-2 | 1B | FineWeb10B | 21.45 | $2.91 \times 10^5$ | $2.33 \times 10^4$ | $2.25 \times 10^3$ | 76312 / 78940 |
| GPT-2 | 60M | FineWeb100B | 35.93 | $2.79 \times 10^3$ | $1.56 \times 10^3$ | $2.35 \times 10^5$ | 14040 / 15712 |
| GPT-2 | 135M | FineWeb100B | 29.58 | $7.68 \times 10^3$ | $2.92 \times 10^3$ | $8.53 \times 10^4$ | 32317 / 32654 |
| GPT-2 | 350M | FineWeb100B | 26.76 | $1.08 \times 10^4$ | $3.24 \times 10^3$ | $6.07 \times 10^4$ | 48063 / 50362 |
| GPT-2 | 1B | FineWeb100B | 21.02 | $2.89 \times 10^5$ | $2.17 \times 10^4$ | $2.27 \times 10^3$ | 76312 / 78940 |
| GPT-2 | 60M | FineWebEdu10B | 32.51 | $2.73 \times 10^3$ | $1.37 \times 10^3$ | $2.40 \times 10^5$ | 14040 / 15712 |
| GPT-2 | 135M | FineWebEdu10B | 23.20 | $7.58 \times 10^3$ | $2.27 \times 10^3$ | $8.65 \times 10^4$ | 32317 / 32654 |
| GPT-2 | 350M | FineWebEdu10B | 21.20 | $1.26 \times 10^4$ | $3.15 \times 10^3$ | $5.20 \times 10^4$ | 48063 / 50362 |
| GPT-2 | 1B | FineWebEdu10B | 18.94 | $2.84 \times 10^5$ | $1.99 \times 10^4$ | $2.31 \times 10^3$ | 76312 / 78940 |
| LLaMA | 60M | FineWeb10B | 38.15 | $2.69 \times 10^3$ | $1.61 \times 10^3$ | $2.44 \times 10^5$ | 13101 / 14548 |
| LLaMA | 135M | FineWeb10B | 34.96 | $5.69 \times 10^3$ | $2.28 \times 10^3$ | $1.15 \times 10^5$ | 22270 / 23894 |
| LLaMA | 350M | FineWeb10B | 33.80 | $1.29 \times 10^4$ | $3.87 \times 10^3$ | $5.08 \times 10^4$ | 48063 / 50362 |
| LLaMA | 1B | FineWeb10B | 22.98 | $2.90 \times 10^5$ | $2.32 \times 10^4$ | $2.26 \times 10^3$ | 76240 / 79012 |
| LLaMA | 60M | FineWeb100B | 34.90 | $2.77 \times 10^3$ | $2.49 \times 10^3$ | $2.37 \times 10^5$ | 13101 / 14548 |
| LLaMA | 135M | FineWeb100B | 34.56 | $5.78 \times 10^3$ | $3.47 \times 10^3$ | $1.13 \times 10^5$ | 22270 / 24014 |
| LLaMA | 350M | FineWeb100B | 32.89 | $1.11 \times 10^4$ | $3.66 \times 10^3$ | $5.90 \times 10^4$ | 48063 / 50362 |
| LLaMA | 1B | FineWeb100B | 21.89 | $2.82 \times 10^5$ | $2.12 \times 10^4$ | $2.32 \times 10^3$ | 76240 / 79012 |
| LLaMA | 60M | FineWebEdu10B | 32.30 | $2.83 \times 10^3$ | $1.13 \times 10^3$ | $2.32 \times 10^5$ | 13101 / 14548 |
| LLaMA | 135M | FineWebEdu10B | 29.51 | $5.68 \times 10^3$ | $1.99 \times 10^3$ | $1.15 \times 10^5$ | 22270 / 23894 |
| LLaMA | 350M | FineWebEdu10B | 28.35 | $1.27 \times 10^4$ | $3.18 \times 10^3$ | $5.16 \times 10^4$ | 48063 / 50362 |
| LLaMA | 1B | FineWebEdu10B | 23.45 | $2.89 \times 10^5$ | $2.02 \times 10^4$ | $2.27 \times 10^3$ | 76240 / 79012 |

Finally, Table 6 presents the algorithmic hyperparameter search grids for all methods, following choices similar to those in (Jordan et al., 2024b; Lau et al., 2025). The grids contain 9 configurations for SGDM, 9 for AdamW, 12 for Muon, and 24 for fixed-rank low-rank Muon. For each reported adaptive-rank run, we use one AdaRank configuration.

For each method, we select the configuration achieving the best validation perplexity at the end of the training budget. The learning rate is warmed up during the first 70% of the training iterations and then follows a

cosine annealing schedule that reduces it to 10% of its initial value. The resulting numerical performance is reported in Section 4.

Table 6: Hyperparameter search spaces and fixed settings for the compared optimizers. Values in braces denote the search grids, whereas individual values denote fixed settings. All methods use a batch size of 2. NS denotes Newton-Schulz.

| Method | Hyperparameter | Search Space or Fixed Setting |
|---|---|---|
| SGDM | Learning rate | $\{10^{-4}, 10^{-3}, 10^{-2}\}$ |
| | Weight decay | $\{0, 10^{-3}, 10^{-2}\}$ |
| AdamW | Learning rate | $\{10^{-4}, 10^{-3}, 10^{-2}\}$ |
| | Weight decay | $\{0, 10^{-3}, 10^{-2}\}$ |
| Muon | Learning rate | $\{10^{-3}, 10^{-2}, 2.5 \times 10^{-2}, 5 \times 10^{-2}\}$ |
| | Number of NS steps | $\{5, 10, 20\}$ |
| | Weight decay | $0$ |
| Fixed-rank Muon | Learning rate | $\{10^{-3}, 10^{-2}, 2.5 \times 10^{-2}, 5 \times 10^{-2}\}$ |
| | Rank | $\{100, 200\}$ |
| | Number of NS steps | $\{5, 10, 20\}$ |
| | Weight decay | $0$ |
| Adaptive-rank Muon | Learning rate for hidden matrices | $2.5 \times 10^{-2}$ |
| | Rank estimator | Stable-rank proxy, updated at every optimizer step |
| | Rank range | $64 \le r_k \le 256$ |
| | Rank-rounding rule | Round upward to the nearest multiple of 32 |
| | Weight decay | $10^{-2}$ |

## B  Proof of the Main Results

In this section, we provide the proofs of our main results presented in Section 3, specifically Theorems 1 to 4.

We first provide three technical lemmas, whose proofs can be found in (He et al., 2025a, Lemmas 1 to 3) and are therefore omitted here. The next lemma provides an expansion for the $\alpha$-power of the Frobenius norm, generalizing the well-known identity $\|U + V\|_F^2 = \|U\|_F^2 + 2\langle U, V\rangle + \|V\|_F^2$ and inequality $\|U + V\|_F^2 \le (1 + c)\|U\|_F^2 + (1 + 1/c)\|V\|_F^2$ for all $U, V \in \mathbb{R}^{m \times n}$ and $c > 0$.

**Lemma 1.** *For all $\alpha \in (1, 2]$, $U, V \in \mathbb{R}^{m \times n}$, and $c > 0$, it holds that*

$$\|U + V\|_F^\alpha \le \|U\|_F^\alpha + \alpha\|U\|_F^{\alpha-2}\langle U, V\rangle + 2\|V\|_F^\alpha, \tag{13}$$

$$\|U + V\|_F^\alpha \le (1 + c)\|U\|_F^\alpha + (2 + (\alpha - 1)^{\alpha-1}c^{1-\alpha})\|V\|_F^\alpha. \tag{14}$$

The next lemma provides an estimation of the partial sums of series.

**Lemma 2.** *Let $\zeta(\cdot)$ be a convex univariate function. Then we have $\sum_{r=a}^{b}\zeta(r) \le \int_{a-1/2}^{b+1/2}\zeta(\tau)\mathrm{d}\tau$ for any integers $a, b$ satisfying $[a - 1/2, b + 1/2] \subset \mathrm{dom}\,\zeta$. Consequently, one has*

$$\sum_{r=a}^{b} r^{-\beta} \le \begin{cases} \ln\left(b + \frac{1}{2}\right) - \ln\left(a - \frac{1}{2}\right) & \text{if } \beta = 1, \\ \frac{\left(b+\frac{1}{2}\right)^{1-\beta} - \left(a-\frac{1}{2}\right)^{1-\beta}}{1-\beta} & \text{if } \beta > 0, \beta \ne 1. \end{cases} \tag{15}$$

We next provide a lemma that will be used to derive complexity bounds for our methods.

**Lemma 3.** *Let $\beta \in (0, 1)$ and $u \in (0, 1/e)$ be given. Then, $v^{-\beta}\ln v \le 2u/\beta$ holds for all $v \ge (u^{-1}\ln(1/u))^{1/\beta}$.*

We next establish a descent property for $f$ along a matrix-signed direction.

**Lemma 4.** *Suppose that Assumption 1 holds. Let $X, M \in \mathbb{R}^{m \times n}$ and $\eta > 0$ be given, and let $X^+ = X - \eta \mathrm{msgn}(M)$. Then we have*

$$f(X^+) \leq f(X) - \eta \|\nabla f(X)\|_* + 2\eta \|\nabla f(X) - M\|_* + \frac{L_* \eta^2}{2}, \tag{16}$$

*where $L_*$ is given in Assumption 1.*

*Proof.* By the definition of the matrix-sign function, one has $\|\mathrm{msgn}(M)\| \leq 1$, and $\langle M, \mathrm{msgn}(M) \rangle = \|M\|_*$. It then follows from these and (4) with $Y = X^+$ that

$$
\begin{aligned}
f(X^+) &\overset{(4)}{\leq} f(X) + \langle \nabla f(X), X^+ - X \rangle + \frac{L_*}{2} \|X^+ - X\|^2 \\
&= f(X) + \langle M, X^+ - X \rangle + \langle \nabla f(X) - M, X^+ - X \rangle + \frac{L_*}{2} \|X^+ - X\|^2 \\
&\leq f(X) - \eta \langle M, \mathrm{msgn}(M) \rangle + \eta \|\nabla f(X) - M\|_* \|\mathrm{msgn}(M)\| + \frac{L_* \eta^2}{2} \|\mathrm{msgn}(M)\|^2 \\
&\leq f(X) - \eta \|M\|_* + \eta \|\nabla f(X) - M\|_* + \frac{L_* \eta^2}{2} \\
&\leq f(X) - \eta \|\nabla f(X)\|_* + 2\eta \|\nabla f(X) - M\|_* + \frac{L_* \eta^2}{2},
\end{aligned}
$$

where the second inequality is due to $X^+ = X - \eta \mathrm{msgn}(M)$ and the trace Hölder inequality, the third inequality is due to $\|\mathrm{msgn}(M)\| \leq 1$ and $\langle M, \mathrm{msgn}(M) \rangle = \|M\|_*$, and the last inequality follows from the triangular inequality. Hence, the conclusion (16) holds as desired. $\square$

### B.1 Proof of the Main Results in Section 3.1

In this subsection, we prove Theorem 1.

***Proof of Theorem 1.*** The proof of (5) can be found in (Halko et al., 2011, Theorem 10.5). Next, we prove $\mathrm{msgn}(QQ^T M) = Q\mathrm{msgn}(Q^T M)$. Let $Q^T M = U_Q \Sigma V^T$ be the reduced SVD of $Q^T M$, where $U_Q$ and $V$ have orthogonal columns, and $\Sigma$ is diagonal. Denote $U = QU_Q$. Since $Q$ and $U_Q$ have orthogonal columns, we obtain that $U$ also has orthogonal columns. Thus, $QQ^T M = U\Sigma V^T$ represents a reduced SVD of $QQ^T M$. Therefore, we have $\mathrm{msgn}(QQ^T M) = UV^T = QU_Q V^T = Q\mathrm{msgn}(Q^T M)$. $\square$

### B.2 Proof of the Main Results in Section 3.2

In this subsection, we give proofs of Theorems 2 and 3.

***Proof of Theorem 2.*** Recall from Algorithm 2 that $M_O^k = \mathrm{msgn}(M_Q^k)$ for all $k \geq 0$. Using this and Lemma 4 with $(X, X^+, M, \eta) = (X^k, X^{k+1}, M_Q^k, \eta_k)$, we obtain that for all $k \geq 0$,

$$f(X^{k+1}) \leq f(X^k) - \eta_k \|\nabla f(X^k)\|_* + 2\eta_k \|\nabla f(X^k) - M_Q^k\|_* + \frac{L_* \eta_k^2}{2}. \tag{17}$$

Summing (17) over $k = 0, \ldots, K-1$ and using $f(X^K) \geq f_{\mathrm{low}}$, we obtain that for all $K \geq 1$,

$$f_{\mathrm{low}} \leq f(X^K) \overset{(17)}{\leq} f(X^0) - \eta_{K-1} \sum_{k=0}^{K-1} \|\nabla f(X^k)\|_* + \sum_{k=0}^{K-1} \left( 2\eta_k \|\nabla f(X^k) - M_Q^k\|_* + \frac{L_* \eta_k^2}{2} \right), \tag{18}$$

where the last inequality is due to (17) and the fact that $\{\eta_k\}$ is nonincreasing. Rearranging this inequality and using $\eta_k = (k+1)^{-1/2}$ for all $k \geq 0$, we obtain that for all $K \geq 3$,

$$\frac{1}{K} \sum_{k=0}^{K-1} \|\nabla f(X^k)\|_* \overset{(18)}{\leq} \frac{f(X^0) - f_{\mathrm{low}}}{K \eta_{K-1}} + \frac{\sum_{k=0}^{K-1} \left( 2\eta_k \|\nabla f(X^k) - M_Q^k\|_* + \frac{L_* \eta_k^2}{2} \right)}{K \eta_{K-1}}$$

$$= \frac{f(X^0) - f_{\text{low}}}{K^{1/2}} + \frac{\sum_{k=0}^{K-1} \left( \frac{2\|\nabla f(X^k) - M_Q^k\|_*}{(k+1)^{1/2}} + \frac{L_*}{2(k+1)} \right)}{K^{1/2}}$$

$$\leq \frac{f(X^0) - f_{\text{low}} + L_* \ln K}{K^{1/2}} + \frac{2\sum_{k=0}^{K-1} \frac{\|\nabla f(X^k) - M_Q^k\|_*}{(k+1)^{1/2}}}{K^{1/2}},$$

where the last inequality follows from $\sum_{k=0}^{K-1} 1/(k+1) \leq \ln(2K+1) \leq 2\ln K$ for all $K \geq 3$ due to (15). Hence, the conclusion (6) holds as desired. $\qquad\square$

***Proof of Theorem 3.*** Using (7), (8), the definition of $\{(\eta_k, \delta_k)\}$, and the same arguments as for proving (6), we obtain that for all $K \geq 3$,

$$\min_{0 \leq k \leq K-1} \{\|\nabla f(X^k)\|\} \leq \frac{(f(X^0) - f_{\text{low}} + L_*) \ln K}{K^{1/2}} + \frac{2}{K^{1/2}} \sum_{k=0}^{K-1} \frac{\delta_k}{(k+1)^{1/2}}$$

$$= \frac{(f(X^0) - f_{\text{low}} + L_*) \ln K}{K^{1/2}} + \frac{2}{K^{1/2}} \sum_{k=0}^{K-1} \frac{1}{k+1}$$

$$\leq \frac{(f(X^0) - f_{\text{low}} + L_* + 2) \ln K}{K^{1/2}} \stackrel{(8)}{=} \frac{U_{\text{gd}} \ln K}{K^{1/2}}, \tag{19}$$

where the last inequality follows from $\sum_{k=0}^{K-1} 1/(k+1) \leq \ln(2K+1) \leq 2\ln K$ for all $K \geq 3$ due to (15). In addition, by Lemma 3 with $(\beta, u, v) = (1/2, \epsilon/(4U_{\text{gd}}), K)$, one can see that

$$K^{-1/2} \ln K \leq \frac{\epsilon}{U_{\text{gd}}} \qquad \forall K \geq \left( \frac{4U_{\text{gd}}}{\epsilon} \ln \left( \frac{4U_{\text{gd}}}{\epsilon} \right) \right)^2,$$

which along with (19) implies that Theorem 3 holds.

$\qquad\square$

## B.3 Proof of the Main Results in Section 3.3

In this subsection, we begin by establishing some technical lemmas and use them to prove Theorems 4. For convenience, we define a sequence of potentials for Algorithm 4 as follows:

$$\mathcal{P}_k := f(X^k) + p_k \|\nabla f(X^k) - M^k\|_F^\alpha \qquad \forall k \geq 0, \tag{20}$$

where the sequence $\{(X^k, M^k)\}$ is generated by Algorithm 4, and $\{p_k\}$ is a sequence of positive scalars that will be specified separately later. The following lemma presents a recurrence relation for the estimation error of the gradient estimators $\{M^k\}$ generated by Algorithm 4.

**Lemma 5.** *Suppose that Assumptions 1 and 2 hold. Let $\{(X^k, M^k)\}$ be generated by Algorithm 4 with input parameters $\{(\eta_k, \theta_k)\}$. Then we have for all $k \geq 0$,*

$$\mathbb{E}_{\xi^{k+1}}[\|M^{k+1} - \nabla f(X^{k+1})\|_F^\alpha] \leq (1 - \theta_k)\|M^k - \nabla f(X^k)\|_F^\alpha + 3L_*^\alpha \theta_k^{1-\alpha} \eta_k^\alpha + 2\sigma^\alpha \theta_k^\alpha, \tag{21}$$

*where $L_*$ is given in Assumption 1, and $\sigma$ and $\alpha$ are given in Assumption 2.*

*Proof.* Fix any $k \geq 0$. It follows from (9) that

$$M^{k+1} - \nabla f(X^{k+1})$$

$$\stackrel{(9)}{=} (1 - \theta_k)(M^k - \nabla f(X^k) + \nabla f(X^k) - \nabla f(X^{k+1})) + \theta_k(G(X^{k+1}; \xi^{k+1}) - \nabla f(X^{k+1})). \tag{22}$$

In addition, we observe from (10) that $M_O^k = \text{msgn}(M_Q^k)$, which along with (11) implies that $\|X^{k+1} - X^k\| = \eta_k\|\text{msgn}(M_Q^k)\| \leq \eta_k$. Also, it follows from [Assumptions 1 and 2](#) that $\mathbb{E}_{\xi^{k+1}}[G(X^{k+1}; \xi^{k+1}) - \nabla f(X^{k+1})] = 0$,

$\mathbb{E}_{\xi^{k+1}}[\|G(X^{k+1};\xi^{k+1}) - \nabla f(X^{k+1})\|_F^\alpha] \leq \sigma^\alpha$, and $\|\nabla f(X^k) - \nabla f(X^{k+1})\|_F \leq \|\nabla f(X^k) - \nabla f(X^{k+1})\|_* \leq L_*\eta_k$. Using these, (13), (14), and (22), we obtain that for all $c > 0$,

$$\mathbb{E}_{\xi^{k+1}}[\|M^{k+1} - \nabla f(X^{k+1})\|_F^\alpha]$$

$$\overset{(22)}{=} \mathbb{E}_{\xi^{k+1}}[\|(1-\theta_k)(M^k - \nabla f(X^k)) + (1-\theta_k)(\nabla f(X^k) - \nabla f(X^{k+1})) + \theta_k(G(X^{k+1};\xi^{k+1}) - \nabla f(X^{k+1}))\|_F^\alpha]$$

$$\overset{(13)}{\leq} \|(1-\theta_k)(M^k - \nabla f(X^k)) + (1-\theta_k)(\nabla f(X^k) - \nabla f(X^{k+1}))\|_F^\alpha$$
$$\quad + 2\mathbb{E}_{\xi^{k+1}}[\|\theta_k(G(X^{k+1};\xi^{k+1}) - \nabla f(X^{k+1}))\|_F^\alpha]$$

$$\overset{(14)}{\leq} (1+c)(1-\theta_k)^\alpha\|M^k - \nabla f(X^k)\|_F^\alpha + L_*^\alpha(2 + (\alpha-1)^{\alpha-1}c^{1-\alpha})(1-\theta_k)^\alpha\eta_k^\alpha + 2\sigma^\alpha\theta_k^\alpha, \tag{23}$$

where the first inequality follows from (13) and $\mathbb{E}_{\xi^{k+1}}[G(X^{k+1};\xi^{k+1})] = \nabla f(X^{k+1})$, the second inequality is due to (14), $\mathbb{E}_{\xi^{k+1}}[\|G(X^{k+1};\xi^{k+1}) - \nabla f(X^{k+1})\|_F^\alpha] \leq \sigma^\alpha$, and $\|\nabla f(X^k) - \nabla f(X^{k+1})\|_F \leq L_*\eta_k$.

When $\theta_k = 1$, (21) clearly holds. For $\theta_k \in (0,1)$, letting $c = (1-\theta_k)^{1-\alpha} - 1$ in (23), and using the fact that $\alpha \in (1,2]$, we have

$$c^{1-\alpha} = ((1-\theta_k)^{1-\alpha} - 1)^{1-\alpha} = \left(\frac{1}{(1-\theta_k)^{\alpha-1}} - 1\right)^{1-\alpha} \leq \left(\frac{1}{1-(\alpha-1)\theta_k} - 1\right)^{1-\alpha} \leq ((\alpha-1)\theta_k)^{1-\alpha},$$

where the first inequality follows from $(1-\tau)^\beta \leq 1 - \beta\tau$ for all $\tau \in (-\infty, 1)$ and $\beta \in [0,1]$. Combining this inequality with (23), one has

$$\mathbb{E}_{\xi^{k+1}}[\|M^{k+1} - \nabla f(X^{k+1})\|_F^\alpha] \leq (1-\theta_k)\|M^k - \nabla f(X^k)\|_F^\alpha + L_*^\alpha(2 + \theta_k^{1-\alpha})(1-\theta_k)^\alpha\eta_k^\alpha + 2\sigma^\alpha\theta_k^\alpha,$$

which along with $\theta_k \in (0,1]$ and $\alpha \in (1,2]$ implies that (21) holds as desired. $\square$

The following lemma establishes a descent property for the potential sequence $\{\mathcal{P}_k\}$ defined below.

**Lemma 6.** *Suppose that Assumptions 1 and 2 hold. Let $\{(X^k, M^k)\}$ be generated by Algorithm 4 with input parameters $\{(\eta_k, \theta_k, \delta_k)\}$, $L_*$ be given in Assumption 1, $\sigma$ and $\alpha$ be given in Assumption 2, and $\varrho := \max\{m,n\}$, and let $\{\mathcal{P}_k\}$ be defined in (20) for $\{(X^k, M^k)\}$ and any nonincreasing positive sequence $\{p_k\}$. Then, it holds that for all $k \geq 0$,*

$$\mathbb{E}_{\xi^{k+1}}[\mathcal{P}_{k+1}] \leq \mathcal{P}_k - \eta_k\|\nabla f(X^k)\|_* + 2\eta_k\delta_k + \frac{L_*\eta_k^2}{2} + \frac{(\alpha-1)(2\varrho^{1/2}\eta_k)^{\alpha/(\alpha-1)}}{\alpha^{\alpha/(\alpha-1)}(\theta_k p_k)^{1/(\alpha-1)}} + 3L_*^\alpha\theta_k^{1-\alpha}\eta_k^\alpha p_k + 2\sigma^\alpha\theta_k^\alpha p_k. \tag{24}$$

*Proof.* Fix any $k \geq 0$. Recall from (10) that $M_O^k = \text{msgn}(M_Q^k)$ and $\|M^k - M_Q^k\|_* \leq \delta_k$. Using these and (16) with $(X^+, X, M, \eta) = (X^{k+1}, X^k, M_Q^k, \eta_k)$, we obtain that

$$f(X^{k+1}) \leq f(X^k) - \eta_k\|\nabla f(X^k)\|_* + 2\eta_k\|\nabla f(X^k) - M_Q^k\|_* + \frac{L_*\eta_k^2}{2}$$

$$\leq f(X^k) - \eta_k\|\nabla f(X^k)\|_* + 2\eta_k\|\nabla f(X^k) - M^k\|_* + 2\eta_k\delta_k + \frac{L_*\eta_k^2}{2}. \tag{25}$$

Combining this with (20) and (21), we obtain that

$$\mathbb{E}_{\xi^{k+1}}[\mathcal{P}_{k+1}] \overset{(20)}{=} \mathbb{E}_{\xi^{k+1}}[f(X^{k+1}) + p_{k+1}\|\nabla f(X^{k+1}) - M^{k+1}\|_F^\alpha]$$

$$\overset{(21)(25)}{\leq} f(X^k) - \eta_k\|\nabla f(X^k)\|_* + 2\eta_k\|\nabla f(X^k) - M^k\|_* + (1-\theta_k)p_{k+1}\|M^k - \nabla f(X^k)\|_F^\alpha$$

$$\quad + 2\eta_k\delta_k + \frac{L_*\eta_k^2}{2} + 3L_*^\alpha\theta_k^{1-\alpha}\eta_k^\alpha p_{k+1} + 2\sigma^\alpha\theta_k^\alpha p_{k+1}$$

$$\leq f(X^k) - \eta_k\|\nabla f(X^k)\|_* + 2\eta_k\varrho^{1/2}\|\nabla f(X^k) - M^k\|_F + (1-\theta_k)p_k\|M^k - \nabla f(X^k)\|_F^\alpha$$

$$+ 2\eta_k\delta_k + \frac{L_*\eta_k^2}{2} + 3L_*^\alpha\theta_k^{1-\alpha}\eta_k^\alpha p_k + 2\sigma^\alpha\theta_k^\alpha p_k, \tag{26}$$

where the second inequality follows from $\|U\|_* \le \varrho^{1/2}\|U\|_F$ for all $U \in \mathbb{R}^{m\times n}$, and the fact that $\{p_k\}$ is nonincreasing. In addition, letting $\alpha' = \alpha/(\alpha-1)$ and using the Young's inequality, we obtain that

$$2\eta_k\varrho^{1/2}\|\nabla f(X^k) - M^k\|_F \le \frac{\left((\alpha\theta_k p_k)^{1/\alpha}\|\nabla f(X^k) - M^k\|_F\right)^\alpha}{\alpha} + \frac{\left(\frac{2\varrho^{1/2}\eta_k}{(\alpha\theta_k p_k)^{1/\alpha}}\right)^{\alpha'}}{\alpha'}$$

$$= \theta_k p_k\|\nabla f(X^k) - M^k\|_F^\alpha + \frac{(\alpha-1)(2\varrho^{1/2}\eta_k)^{\alpha/(\alpha-1)}}{\alpha^{\alpha/(\alpha-1)}(\theta_k p_k)^{1/(\alpha-1)}}.$$

This along with (26) implies that

$$\mathbb{E}_{\xi^{k+1}}[\mathcal{P}_{k+1}] \le f(X^k) + p_k\|\nabla f(X^k) - M^k\|_F^\alpha - \eta_k\|\nabla f(X^k)\|_*$$

$$+ 2\eta_k\delta_k + \frac{L_*\eta_k^2}{2} + \frac{(\alpha-1)(2\varrho^{1/2}\eta_k)^{\alpha/(\alpha-1)}}{\alpha^{\alpha/(\alpha-1)}(\theta_k p_k)^{1/(\alpha-1)}} + 3L_*^\alpha\theta_k^{1-\alpha}\eta_k^\alpha p_k + 2\sigma^\alpha\theta_k^\alpha p_k.$$

The conclusion (24) then follows from this and (20). □

We are now ready to prove Theorem 4.

***Proof of Theorem 4.*** Let $\{(X^k, M^k)\}$ be generated by Algorithm 4 with the definition of $\{(\eta_k, \theta_k)\}$, and $\{\mathcal{P}_k\}$ be defined in (20) with such $\{(X^k, M^k)\}$ and the following $\{p_k\}$:

$$p_k = (k+1)^{(\alpha^2-3\alpha+2)/(3\alpha-2)} \qquad \forall k \ge 0. \tag{27}$$

Since $\alpha \in (1,2]$, one can see that $\{p_k\}$ is nonincreasing. Also, observe from the definition of $\{(\eta_k, \theta_k)\}$ that $\{\eta_k\} \subset (0,1]$ and $\{\theta_k\} \subset (0,1]$. Hence, $\{(\eta_k, \theta_k, p_k)\}$ satisfies the assumptions in Lemma 6 and Algorithm 4. In addition, by (20) and (27), one has that

$$\mathbb{E}[\mathcal{P}_0] = f(X^0) + \mathbb{E}[\|G(X^0;\xi^0) - \nabla f(X^0)\|_F^\alpha] \le f(X^0) + \sigma^\alpha, \tag{28}$$

$$\mathbb{E}[\mathcal{P}_K] \ge \mathbb{E}[f(X^K)] \ge f_{\text{low}}. \tag{29}$$

Taking expectation on both sides of (24) with respect to $\{\xi^i\}_{i=0}^{k+1}$, we have for all $k \ge 0$,

$$\mathbb{E}[\mathcal{P}_{k+1}] \le \mathbb{E}[\mathcal{P}_k] - \eta_k\mathbb{E}[\|\nabla f(X^k)\|_*] + 2\eta_k\delta_k + \frac{L_*\eta_k^2}{2} + \frac{(\alpha-1)(2\varrho^{1/2}\eta_k)^{\alpha/(\alpha-1)}}{\alpha^{\alpha/(\alpha-1)}(\theta_k p_k)^{1/(\alpha-1)}}$$

$$+ 3L_*^\alpha\theta_k^{1-\alpha}\eta_k^\alpha p_k + 2\sigma^\alpha\theta_k^\alpha p_k.$$

Summing up this inequality over $k = 0, \ldots, K-1$, and using (28) and (29), we obtain that for all $K \ge 1$,

$$f_{\text{low}} \overset{(29)}{\le} \mathbb{E}[\mathcal{P}_K] \le \mathbb{E}[\mathcal{P}_0] - \sum_{k=0}^{K-1}\eta_k\mathbb{E}[\|\nabla f(X^k)\|_*]$$

$$+ \sum_{k=0}^{K-1}\left(2\eta_k\delta_k + \frac{L_*\eta_k^2}{2} + \frac{(\alpha-1)(2\varrho^{1/2}\eta_k)^{\alpha/(\alpha-1)}}{\alpha^{\alpha/(\alpha-1)}(\theta_k p_k)^{1/(\alpha-1)}} + 3L_*^\alpha\theta_k^{1-\alpha}\eta_k^\alpha p_k + 2\sigma^\alpha\theta_k^\alpha p_k\right) \tag{30}$$

$$\overset{(28)}{\le} f(X^0) + \sigma^\alpha - \eta_{K-1}\sum_{k=0}^{K-1}\mathbb{E}[\|\nabla f(X^k)\|_*]$$

$$+ \sum_{k=0}^{K-1}\left(2\eta_k\delta_k + \frac{L_*\eta_k^2}{2} + \frac{(\alpha-1)(2\varrho^{1/2}\eta_k)^{\alpha/(\alpha-1)}}{\alpha^{\alpha/(\alpha-1)}(\theta_k p_k)^{1/(\alpha-1)}} + 3L_*^\alpha\theta_k^{1-\alpha}\eta_k^\alpha p_k + 2\sigma^\alpha\theta_k^\alpha p_k\right), \tag{31}$$

where the last inequality follows from (28) and the fact that $\{\eta_k\}$ is nonincreasing. Rearranging the terms in (31), and using the definition of $\{(\eta_k, \theta_k, \delta_k)\}$, (12), and (27), we obtain that for all $K \geq 3$,

$$
\begin{aligned}
\frac{1}{K} \sum_{k=0}^{K-1} \mathbb{E}[\|\nabla f(X^k)\|_*] & \overset{(31)}{\leq} \frac{f(X^0) - f_{\text{low}} + \sigma^\alpha}{K\eta_{K-1}} \\
& + \frac{\sum_{k=0}^{K-1}\left(2\eta_k\delta_k + \frac{L_*\eta_k^2}{2} + \frac{(\alpha-1)(2\varrho^{1/2}/\alpha)^{\alpha/(\alpha-1)}\eta_k^{\alpha/(\alpha-1)}}{(\theta_k p_k)^{1/(\alpha-1)}}\right)}{K\eta_{K-1}} + \frac{\sum_{k=0}^{K-1}\left(3L_*^\alpha\theta_k^{1-\alpha}\eta_k^\alpha p_k + 2\sigma^\alpha\theta_k^\alpha p_k\right)}{K\eta_{K-1}} \\
& \overset{(27)}{=} \frac{f(X^0) - f_{\text{low}} + \sigma^\alpha}{K^{(\alpha-1)/(3\alpha-2)}} + \frac{\sum_{k=0}^{K-1}\frac{L_*}{2(k+1)^{2(2\alpha-1)/(3\alpha-2)}}}{K^{(\alpha-1)/(3\alpha-2)}} + \frac{\sum_{k=0}^{K-1}\frac{2+(\alpha-1)(2\varrho^{1/2}/\alpha)^{\alpha/(\alpha-1)}+3L_*^\alpha+2\sigma^\alpha}{k+1}}{K^{(\alpha-1)/(3\alpha-2)}} \\
& \overset{(12)}{\leq} \frac{U_{\text{mn}}\ln K}{K^{(\alpha-1)/(3\alpha-2)}},
\end{aligned}
$$

where the second inequality follows from $\sum_{k=0}^{K-1} 1/(k+1) \leq 2\ln K$ due to (15) and $K \geq 3$, and $\sum_{k=0}^{K-1} 1/(k+1)^{2(2\alpha-1)/(3\alpha-2)} \leq (3\alpha-2)2^{\alpha/(3\alpha-2)}/\alpha < 4$ due to (15) and $(3\alpha-2)/\alpha \in (1, 2]$. Recall that $\iota_K$ is uniformly selected from $\{0, \ldots, K-1\}$. It then follows from this and the above inequality that for all $K \geq 3$,

$$
\mathbb{E}[\|\nabla f(X^{\iota_K})\|_*] \leq \frac{U_{\text{mn}}\ln K}{K^{(\alpha-1)/(3\alpha-2)}}. \tag{32}
$$

In addition, by Lemma 3 with $(\beta, u, v) = ((\alpha-1)/(3\alpha-2), (\alpha-1)\epsilon/(2(3\alpha-2)U_{\text{mn}}), K)$, one can see that

$$
K^{-(\alpha-1)/(3\alpha-2)}\ln K \leq \frac{\epsilon}{U_{\text{mn}}} \qquad \forall K \geq \left(\frac{2(3\alpha-2)U_{\text{mn}}}{(\alpha-1)\epsilon}\ln\left(\frac{2(3\alpha-2)U_{\text{mn}}}{(\alpha-1)\epsilon}\right)\right)^{(3\alpha-2)/(\alpha-1)},
$$

which together with (32) implies that Theorem 4 holds. $\qquad\square$

