# OpenReview forum: "Low-rank orthogonalization for large-scale matrix optimization with applications to foundation model training"
_TMLR — Under review for TMLR_

### Review · Reviewer_8VeZ · 2026-05-16

**Summary Of Contributions:**

This paper proposes a low-rank version of matrix orthogonalization for Muon-style optimizers. The idea is simple: instead of orthogonalizing the full gradient or momentum matrix, the method first sketches it, builds a low-rank approximation, and then applies the matrix-sign update on that smaller object. The paper applies this idea to low-rank MSGD and low-rank Muon, gives convergence results, and tests the method on GPT-2 and LLaMA continued-training experiments. The results are promising: low-rank Muon often improves over vanilla Muon on larger models and can reduce some of the optimizer overhead.

The main strength of the paper is that the idea is easy to understand and potentially useful for large-scale training. My main concern is that the empirical and theoretical evidence does not yet fully support the strongest claims made in the paper.

**Audience:**

Yes

**Audience Explanation:**

Muon-style optimizers are receiving a lot of attention, and a cheaper low-rank orthogonalization step is a natural and useful direction.

**Claims And Evidence:**

No

**Claims Explanation:**

The experiments are interesting, but I think the current evidence is not strong enough for the paper’s broad claims. Low-rank Muon does improve over vanilla Muon in many larger GPT-2 and LLaMA settings, but the gains are not uniform, especially for smaller models. Also, all Muon-style methods are still slower than AdamW or SGDM in many settings, so the efficiency claim needs to be stated more carefully.

I also have concerns about reproducibility. The paper says the hyperparameters were carefully tuned, but the main text does not give enough detail about the tuning budget, learning-rate schedules, batch sizes, token counts, validation setup, or number of random seeds. Since the results are fairly sensitive, this information is important.

There is also a gap between the theory and the experiments. The theory assumes a safeguarded low-rank approximation with a controlled residual error, while the experiments mostly use fixed ranks such as 100 and 200. The paper does not show that the required residual conditions actually hold in the training runs.

Finally, some table entries look suspicious. For example, in the LLaMA results, some methods have exactly identical perplexities and times across several settings. These may be correct, but they need to be checked and explained.

**Requested Changes:**

1. Clarify the gap between the theory and the implementation. The analysis uses controlled low-rank approximation errors, but the experiments use fixed ranks. The authors should either justify this choice empirically or make clear that the theory does not directly cover the practical implementation.
2. Add more experimental details. Please include the full hyperparameter search ranges, tuning budget, batch size, token budget, learning-rate schedules, validation setup, hardware details, and number of seeds.
3. Report uncertainty. At least some key experiments should be repeated with multiple seeds, or the authors should provide another way to show that the improvements are stable.
4. Check the table values. Some LLaMA results appear duplicated across methods, which raises concern about copy-paste or logging errors.
5. Clarify whether the experiments are really “pretraining.” The paper says the models are initialized from pretrained GPT-2 and LLaMA weights, so “continued pretraining” seems more accurate.
6. The paper would be stronger with time-to-target perplexity, tokens/sec, optimizer overhead, and peak memory. It would also help to show singular-value spectra or effective ranks of real gradients during training, since the method is motivated by low-rank gradient structure. More rank ablations beyond 100 and 200 would also be useful.

---

> ### Author Response · Authors · 2026-06-28
> **Thank you for your insightful comments!**
>
> > This paper proposes a low-rank version of matrix orthogonalization for Muon-style optimizers. The idea is simple: instead of orthogonalizing the full gradient or momentum matrix, the method first sketches it, builds a low-rank approximation, and then applies the matrix-sign update on that smaller object. The paper applies this idea to low-rank MSGD and low-rank Muon, gives convergence results, and tests the method on GPT-2 and LLaMA continued-training experiments. The results are promising: low-rank Muon often improves over vanilla Muon on larger models and can reduce some of the optimizer overhead.
>
> > The main strength of the paper is that the idea is easy to understand and potentially useful for large-scale training.
>
> Thank you for your encouraging comments!
>
> > My main concern is that the empirical and theoretical evidence does not yet fully support the strongest claims made in the paper.
>
> Below, we provide justifications, moderate our claims, and address your concerns one by one.
>
> > The experiments are interesting, but I think the current evidence is not strong enough for the paper’s broad claims. Low-rank Muon does improve over vanilla Muon in many larger GPT-2 and LLaMA settings, but the gains are not uniform, especially for smaller models. Also, all Muon-style methods are still slower than AdamW or SGDM in many settings, so the efficiency claim needs to be stated more carefully.
>
> We will tone down our efficiency claims and clarify that low-rank Muon primarily improves upon vanilla Muon for larger GPT-2 and LLaMA models.
>
> > I also have concerns about reproducibility. The paper says the hyperparameters were carefully tuned, but the main text does not give enough detail about the tuning budget, learning-rate schedules, batch sizes, token counts, validation setup, or number of random seeds. Since the results are fairly sensitive, this information is important.
>
> We provide additional details on the numerical experiments in Appendix A, including the model configurations, batch size, token budget, training duration, validation setup and metric, learning-rate schedule, hardware, number of random seeds, and tuning budget. Specifically, each run uses a batch size of 2, at most 131,072 training tokens per iteration, 5,000 optimizer iterations, a held-out validation set containing 10,485,760 tokens, a warm-up phase covering the first 70% of training followed by cosine annealing to 10% of the initial learning rate, and two NVIDIA A100 80 GB GPUs. In the comparison tables (Tables 4 and 5), we report the empirical variability across three runs with different random seeds. We also provide the hyperparameter search grids for all methods in Table 6.
>
> > There is also a gap between the theory and the experiments. The theory assumes a safeguarded low-rank approximation with a controlled residual error, while the experiments mostly use fixed ranks such as 100 and 200. The paper does not show that the required residual conditions actually hold in the training runs.
>
> Thank you for bringing this up. We provide numerical verification in Figs. 5 and 8, showing that the nuclear-norm error remains bounded at the order $\mathcal{O}(k^{-1/2})$ during GPT-2 and LLaMA training.
>
> > Finally, some table entries look suspicious. For example, in the LLaMA results, some methods have exactly identical perplexities and times across several settings. These may be correct, but they need to be checked and explained.
>
> We apologize for the oversight. During the preparation of the tables, two rows were inadvertently duplicated due to a copy-and-paste error and were not properly checked. We have corrected these entries in the revised manuscript and carefully cross-checked all reported results against the original experimental records to ensure their accuracy.
>
> > Muon-style optimizers are receiving a lot of attention, and a cheaper low-rank orthogonalization step is a natural and useful direction.
>
> Thank you for your encouraging comments. Below, we address your requested changes.
>
> > 1. Clarify the gap between the theory and the implementation. The analysis uses controlled low-rank approximation errors, but the experiments use fixed ranks. The authors should either justify this choice empirically or make clear that the theory does not directly cover the practical implementation.
>
> We now include numerical experiments verifying that the nuclear-norm error is bounded by a sequence of order $\mathcal{O}(k^{-1/2})$ during GPT-2 and LLaMA training.
>
> > 2. Add more experimental details. Please include the full hyperparameter search ranges, tuning budget, batch size, token budget, learning-rate schedules, validation setup, hardware details, and number of seeds.
>
> We now include the experimental details in Appendix A.

---

> ### Author Response · Authors · 2026-06-28
> **Additional responses**
>
> > 3. Report uncertainty. At least some key experiments should be repeated with multiple seeds, or the authors should provide another way to show that the improvements are stable.
>
> We now report empirical multi-run variability in the main GPT-2 and LLaMA comparison tables. Specifically, the $\pm$ values in Tables 1 and 2 summarize variation over three runs with different random seeds for the reported cells.
>
> > 4. Check the table values. Some LLaMA results appear duplicated across methods, which raises concern about copy-paste or logging errors.
>
> We apologize for the oversight. We have corrected these entries in the revised manuscript and carefully cross-checked all reported results against the original experimental records.
>
> > 5. Clarify whether the experiments are really “pretraining.” The paper says the models are initialized from pretrained GPT-2 and LLaMA weights, so “continued pretraining” seems more accurate.
>
> Thank you for pointing this out! We would like to clarify that our experiments on GPT-2 and LLaMA models are randomly initialized and do not use pretrained checkpoints. We have carefully reviewed the manuscript and revised the relevant text accordingly.
>
> > 6. (a) The paper would be stronger with time-to-target perplexity, tokens/sec, optimizer overhead, and peak memory.
>
> We now report these metrics in Tables 4 and 5 of Appendix A.
>
> > 6. (b) It would also help to show singular-value spectra or effective ranks of real gradients during training, since the method is motivated by low-rank gradient structure.
>
> We now plot the leading singular-value spectra of the Q/K/V gradients associated with a neural network block over the first 400 iterations of GPT-2 and LLaMA pretraining in Figs. 6 and 9.
>
> > 6. (c) More rank ablations beyond 100 and 200 would also be useful.
>
> To address the choice of rank, we have added an adaptive-rank variant of low-rank Muon that dynamically selects the rank based on the stable rank, a numerically computable proxy for matrix rank. This variant achieves performance comparable to that of the fixed-rank variants with ranks 100 and 200, while reducing the need for hyperparameter tuning. We have added the corresponding numerical results to Sec. 4.

---

### Review · Reviewer_UHsh · 2026-05-24

**Summary Of Contributions:**

This paper incorporates the low-rank matrix sign (i.e., matrix sign of a low-rank approximate matrix) in the matrix-variate gradient descent optimizer. The key idea is the low-rank orthogonalization method based on Gaussian sketch, which significantly reduces the computational cost of matrix sign under high-dimensional settings. The paper claims that the proposed method is more robust to noise than Newton–Schulz iterations. Convergence of the proposed algorithm is established, and empirical comparisons are conducted against Moun, Adam, SGDM and etc.

I have the following comments
1.  I fail to understand the numerical experiment which aims to claim the noise robustness of algorithm 1. What do you mean by "the trace of the empirical covariance matrix of the estimates of msgn(MN)"? $tr(M_o^TM_o)$? In Figure 3, the legend “RS” should be "GS". Why the x-axis in Figure 3 ranges from 0.001 to 1, while the statement says $\sigma^2\in (0.1,1,10 )$?

2. In algorithms 3 and 4, how is the rank $r_k$ determined? By naive try-and-error search? In the experiment, the authors try Muon with 100 or 200 rank. Does it mean that the rank is fixed rather than adaptive as described by Algorithm 4?

3. Typo: Proof of Lemme 5, 2 lines below equation 22: "Also, it follows from Assumption 1 that" --> "Also, it follows from Assumptions 1 and 2 that"

4. The last line on page, def of  SNSP: What does this expectation mean here? There is no randomness in the equation

5. Can authors discuss the relationship and difference between the existing convergence of Muon and this paper?

6. The abstract of the paper says "low-rank Muon achieving promising results in GPT-2 and LLaMA pretraining". But the experiments in the paper is really about model finetuning (i.e., the models are initialized from pre-trained checkpoint). A recent ICLR paper combines Muon with low rank LoRA finetuning "LoRA meets Riemannion: Muon Optimizer for Parametrization-independent Low-Rank Adapters". I think a comparison with this work will be helpful.

**Additional Comments:**

no

**Audience:**

Yes

**Audience Explanation:**

The paper investigates how to combine low-rank approximation with the matrix-variate optimizer. I personally think this is a very interesting direction and audience in LLM/DNN optimizations will be inerested in this topic.

**Broader Impact Concerns:**

No concern.

**Claims And Evidence:**

No

**Claims Explanation:**

The experiments are all about the model finetuning, not pre-training. Also, I think the experiment scale is kind of small and the improvement is marginal under the given model size.

**Requested Changes:**

Please refer to my comments.

---

> ### Author Response · Authors · 2026-06-27
> **Thank you for your insightful comments!**
>
> > **Summary Of Contributions:** This paper incorporates the low-rank matrix sign (i.e., matrix sign of a low-rank approximate matrix) in the matrix-variate gradient descent optimizer. The key idea is the low-rank orthogonalization method based on Gaussian sketch, which significantly reduces the computational cost of matrix sign under high-dimensional settings. The paper claims that the proposed method is more robust to noise than Newton–Schulz iterations. Convergence of the proposed algorithm is established, and empirical comparisons are conducted against Moun, Adam, SGDM and etc.
>
> Thank you for your encouraging comments!
>
> > I have the following comments
>
> Below, we address your comments one by one.
>
> > 1. (a) I fail to understand the numerical experiment which aims to claim the noise robustness of algorithm 1. What do you mean by "the trace of the empirical covariance matrix of the estimates of msgn(MN)"? $tr(M_o^TM_o)$?
>
> For each $M$, we first calculate $50$ estimates of $\mathrm{msgn}(M)$ in the presence of noise, denoted by {$\mathrm{msgn}(M_{N_i})$}, $i=1,\ldots,50$, where {$N_i$} denotes the set of stimulated noise. To characterize the variability of these estimates, we vectorize them and compute their empirical covariance matrix. We then use the trace of this empirical covariance matrix to quantify the overall variability of the estimates. We provide an explanation of this procedure in the footnote.
>
> > 1. (b) In Figure 3, the legend “RS” should be "GS". Why the x-axis in Figure 3 ranges from 0.001 to 1, while the statement says $\sigma^2\in(0.1,1,10)$?
>
> Thank you for catching these errors. They have now been fixed.
>
> > 2. (a) In algorithms 3 and 4, how is the rank $r_k$ determined? By naive try-and-error search?
>
> We use the stable rank, a numerical proxy for rank, as a reference and tune the input rank based on the validation perplexity.
>
> In addition, to investigate the effect of varying the rank, we have added an adaptive-rank variant of low-rank Muon that dynamically selects the rank based on the stable rank of {$M^k$}. This variant achieves performance comparable to that of the fixed-rank versions with ranks of 100 and 200, while reducing the effort required for hyperparameter tuning. We add numerical results for the adaptive-rank variant to Sec. 4.
>
> > 2. (b) In the experiment, the authors try Muon with 100 or 200 rank. Does it mean that the rank is fixed rather than adaptive as described by Algorithm 4?
>
> We describe Algorithms 3 and 4 so that $r_k$ may vary across iterations, allowing more flexibility in rank selection while remaining within our analytical framework. In the numerical experiments, the fixed-rank version of Muon serves as a simple implementation and is a special case covered by our algorithmic framework. An adaptive-rank variant is now added to Sec. 4.
>
> > 3. Typo: Proof of Lemme 5, 2 lines below equation 22: "Also, it follows from Assumption 1 that" --> "Also, it follows from Assumptions 1 and 2 that"
>
> Fixed. Thank you!
>
> > 4. The last line on page, def of SNSP: What does this expectation mean here? There is no randomness in the equation
>
> The expectation is taken over all randomness in the algorithm, including the sampling randomness and the randomness involved in selecting a solution. We have clarified this in the revised manuscript.
>
> > 5. Can authors discuss the relationship and difference between the existing convergence of Muon and this paper?
>
> We allow inexact computation of the matrix sign with errors bounded in the nuclear norm. In addition, we study convergence under heavy-tailed noise. Both aspects distinguish our analysis from existing convergence analyses of vanilla Muon. We will add a discussion to introduction.
>
> > 6. The abstract of the paper says "low-rank Muon achieving promising results in GPT-2 and LLaMA pretraining". But the experiments in the paper is really about model finetuning (i.e., the models are initialized from pre-trained checkpoint). A recent ICLR paper combines Muon with low rank LoRA finetuning "LoRA meets Riemannion: Muon Optimizer for Parametrization-independent Low-Rank Adapters". I think a comparison with this work will be helpful.
>
> Thank you for pointing this out! We would like to clarify that our experiments on GPT-2 and LLaMA models are randomly initialized and do not use pretrained checkpoints. We have carefully reviewed the manuscript and revised the relevant text accordingly. We have also added the paper “LoRA Meets Riemannian: Muon Optimizer for Parametrization-Independent Low-Rank Adapters” as a reference on low-rank variants of Muon for fine-tuning.

---

> > ### Author Response · Authors · 2026-06-27
> > **Additional responses**
> >
> > > The experiments are all about the model finetuning, not pre-training. Also, I think the experiment scale is kind of small and the improvement is marginal under the given model size.
> >
> > We apologize for the confusion caused by the previous version of the manuscript. We would like to clarify that the experiments on both GPT-2 and LLaMA are conducted from scratch: all models are randomly initialized, and no pretrained checkpoints are used. Therefore, the reported results correspond to pre-training rather than fine-tuning.
> >
> > Regarding the experimental scale, due to computational resource limitations, the current model sizes represent the largest scale that we are able to evaluate. Nevertheless, we believe that the observed improvements, although modest, consistently demonstrate the effectiveness of the proposed method under the considered settings.
> >
> > > The paper investigates how to combine low-rank approximation with the matrix-variate optimizer. I personally think this is a very interesting direction and audience in LLM/DNN optimizations will be inerested in this topic.
> >
> > Thank you for your encouraging comments!

---

### Review · Reviewer_5Ten · 2026-06-16

**Summary Of Contributions:**

The paper develops a low-rank orthogonalization method using sketching techniques. Expanding on this, the paper further proposes low rank matrix-signed gradient descent with and without adaptive ranks, as well as low-rank Muon. The authors provide convergence rate guarantees on their  low-rank matrix-signed gradient descent methods on bounded, smooth problems. Additionally, they provide GPT-2 and LLaMA pre-training results comparing their low-rank Muon with standard optimizers, i.e., SGD, Adam, SOAP, and vanilla Muon.

**Additional Comments:**

At least one of the authors seems to have revealed their identity by providing non-anonymous Github links in both the abstract and Section 4, which violates TMLR's double-blind policy.

**Audience:**

Yes

**Audience Explanation:**

Since it was first released, Muon has gained widespread attention in the AI and optimization literature. Despite Muon's success in LLM pre-training, applying the Newton-Schulz step to many LLM weight matrices can make each Muon step computationally expensive compared to other optimizer steps. This work proposes an effective method to make each Muon step more efficient while being competitive with original Muon's performance, which I believe will be of interest to the community.

**Broader Impact Concerns:**

None, to my knowledge.

**Claims And Evidence:**

Yes

**Claims Explanation:**

The authors claim their low-rank Muon achieves competitive results in GPT-2 and LLaMA pre-training, and leads to significant computational savings over full-rank Muon at larger model scales. Tables 1 and 2 support this claim in my opinion, as it shows low rank Muon achieves similar validation perplexity to Muon, with the difference in computational time growing as the model size increases. Furthermore, low-rank Muon still out-performs the other non-Muon optimizers. Finally, the authors provide proofs for their theoretical results, which seem correct to me. However, I did not go through the proofs in detail line-by-line.

**Requested Changes:**

I believe the following changes would strengthen the work, but are not critical to recommending acceptance / rejection.

1. In the Section 3.1, add experiments measuring the distance / alignment between the estimated and ground truth $\operatorname{msgn}(M_N)$. The current results show that compared to vanilla Newton-Schulz, the matrix sign estimates from low rank orthogonalization is more computationally efficient and has lower total variance. However, these experiments do not indicate if low rank orthogonalization performs better than Newton Schulz at estimating the ground truth $\operatorname{msgn}(\cdot)$ in the first place.

2. In Sections 3.2 and 3.3, add discussions providing insight into necessary / sufficient per-iteration ranks $r_k$ needed to achieve the assumed tolerances $\delta_k$ in Theorems 3 and 4. Those results assume the $\delta_k$'s decrease at a $\mathcal{O}\left( 1 / \operatorname{poly}(k) \right)$ rate, i.e., the low-rank gradient (Theorem 3) or momentum (Theorem 4) estimates must get better after each iteration. This seems like a strong assumption, as it may be the case that very high ranks must be needed to achieve these target tolerances even at moderate $k$. Discussing necessary / sufficient per-iteration ranks $r_k$ would clarify how low of a rank can these methods tolerate in theory.

3. Add an experiment on the cost-performance tradeoff in LLM pretraining by varying the chosen rank. The authors' current results show performance at two fixed ranks for low-rank Muon. It is unclear what the tradeoff is between computational complexity and LLM performance as the chosen rank varies.

---

> ### Author Response · Authors · 2026-06-27
> **Thank you for your insightful comments!**
>
> > **Summary Of Contributions:** The paper develops a low-rank orthogonalization method using sketching techniques. Expanding on this, the paper further proposes low rank matrix-signed gradient descent with and without adaptive ranks, as well as low-rank Muon. The authors provide convergence rate guarantees on their low-rank matrix-signed gradient descent methods on bounded, smooth problems. Additionally, they provide GPT-2 and LLaMA pre-training results comparing their low-rank Muon with standard optimizers, i.e., SGD, Adam, SOAP, and vanilla Muon.
>
> >The authors claim their low-rank Muon achieves competitive results in GPT-2 and LLaMA pre-training, and leads to significant computational savings over full-rank Muon at larger model scales. Tables 1 and 2 support this claim in my opinion, as it shows low rank Muon achieves similar validation perplexity to Muon, with the difference in computational time growing as the model size increases. Furthermore, low-rank Muon still out-performs the other non-Muon optimizers. Finally, the authors provide proofs for their theoretical results, which seem correct to me. However, I did not go through the proofs in detail line-by-line.
>
> > Since it was first released, Muon has gained widespread attention in the AI and optimization literature. Despite Muon's success in LLM pre-training, applying the Newton-Schulz step to many LLM weight matrices can make each Muon step computationally expensive compared to other optimizer steps. This work proposes an effective method to make each Muon step more efficient while being competitive with original Muon's performance, which I believe will be of interest to the community.
>
> Thank you for your encouraging comments!
>
> > **Requested Changes:** I believe the following changes would strengthen the work, but are not critical to recommending acceptance / rejection.
>
> Below, we address your requested changes one by one.
>
> > 1. In the Section 3.1, add experiments measuring the distance / alignment between the estimated and ground truth $\mathrm{msgn}(M_N)$. The current results show that compared to vanilla Newton-Schulz, the matrix sign estimates from low rank orthogonalization is more computationally efficient and has lower total variance. However, these experiments do not indicate if low rank orthogonalization performs better than Newton Schulz at estimating the ground truth $\mathrm{msgn}(\cdot)$ in the first place.
>
> We now add experiments in Sec. 3.1 comparing our low-rank method with the full-rank Newton-Schulz iteration for estimating the ground-truth matrix sign operator in the presence of noise. The Frobenius and alignment errors plotted in Fig. 4 illustrate the trade-off between estimation accuracy and input rank. We observe that, for matrices with certain low-rank structures, our low-rank method estimates the ground truth significantly more accurately than the full-rank Newton-Schulz iteration.
>
> > 2. (a) In Sections 3.2 and 3.3, add discussions providing insight into necessary / sufficient per-iteration ranks $r_k$ needed to achieve the assumed tolerances $\delta_k$ in Theorems 3 and 4. Those results assume the $\delta_k$'s decrease at a $\mathcal{O}(1/\mathrm{ploy}(k))$ rate, i.e., the low-rank gradient (Theorem 3) or momentum (Theorem 4) estimates must get better after each iteration. This seems like a strong assumption, as it may be the case that very high ranks must be needed to achieve these target tolerances even at moderate $k$.
>
> Thank you for pointing this out. To establish a convergence guarantee for the low-rank variant of Muon applied to **general matrix optimization problems**, we assume that the sequence of low-rank estimation errors, {$\||M^k-M_O^k\||_*$}, is bounded by a diminishing sequence {$\delta_k$}. We would like to provide the following additional justifications for this condition:
>
> - For GPT-2 and LLaMA training, this condition on {$||M^k-M_O^k||_*$} is not restrictive because of the benign low-rank structure of the gradients. We numerically verify this condition by visualizing the estimation errors for the gradients associated with the Q, K, and V matrices in a particular block during GPT-2 and LLaMA pretraining. Our results indicate that when the input rank is set to 100 or 200, these estimation errors are bounded by $\mathcal{O}(k^{-1/2})$, the most stringent decay rate required in Theorems 3 and 4. We now present the corresponding plots in Figs. 5 and 8.
>
> - For convergence alone, the dominating sequence {$\delta_k$} need not decay at the rate specified in Theorems 3 and 4. As long as {$\delta_k$} decays polynomially, we can establish the stationarity guarantee
> $$\frac{1}{K}\sum_{k=0}^{K-1}\mathbb{E}[\||\nabla f(X^k)\||_*]\to 0\quad\text{as}\quad K\to\infty.$$
> We choose a particular polynomially decaying sequence {$\delta_k$} in Theorems 3 and 4 only for ease of presentation and to preserve the optimal convergence rate. We have now added discussions to Remark 4.

---

> ### Author Response · Authors · 2026-06-27
> **Additional responses**
>
> > 2. (b)  Discussing necessary / sufficient per-iteration ranks would clarify how low of a rank $r_k$ can these methods tolerate in theory.
>
> The smallest suitable input rank for implementing low-rank Muon depends on the intrinsic rank of the neural network’s weight gradients. Our theoretical framework is developed for general matrix optimization problems and does not account for the low-rank structure of LLM training problems. Therefore, it does not provide a rigorous lower bound on the rank required to ensure accurate estimation. In our experiments, we selected ranks of 100 and 200 through hyperparameter tuning.
>
> > 3. Add an experiment on the cost-performance tradeoff in LLM pretraining by varying the chosen rank. The authors' current results show performance at two fixed ranks for low-rank Muon. It is unclear what the tradeoff is between computational complexity and LLM performance as the chosen rank varies.
>
> A systematic study of the cost-performance trade-off across different rank choices would be interesting. However, this trade-off depends heavily on the model size and training configuration, and a comprehensive rank sweep would require multiple additional LLM pretraining runs. We therefore believe that such a study is beyond the scope of this paper.
>
> Nevertheless, to investigate the effect of varying the rank, we have added an adaptive-rank variant of low-rank Muon that dynamically selects the rank based on the stable rank of {$M^k$}. This variant achieves performance comparable to that of the fixed-rank versions with ranks of 100 and 200, while reducing the effort required for hyperparameter tuning. We add numerical results for the adaptive-rank variant to Sec. 4.
>
> > **Additional comments:** At least one of the authors seems to have revealed their identity by providing non-anonymous Github links in both the abstract and Section 4, which violates TMLR's double-blind policy.
>
> We appreciate the reviewer’s kind reminder. We sincerely apologize for our careless mistake in including identifiable author information in the GitHub link. We have now removed this identifying information from the revised blinded manuscript.

---

### Author Response · Authors · 2026-06-30
**The revised manuscript uploaded**

Thank you for taking the time and effort to review our paper and provide insightful comments. We have revised our manuscript according to your suggestions. The revised version has now been uploaded, with major changes highlighted in blue.